



# Multi-year precipitation characteristics based on in-situ and remote sensing observations at Ny-Ålesund, Svalbard

Kerstin Ebell[1], Christian Buhren[1], Rosa Gierens[1], Giovanni Chellini[1], Melanie Lauer[1], Andreas Walbröl[1], Sandro Dahlke[2], Pavel Krobot[1], and Mario Mech[1]

[1]Institute for Geophysics and Meteorology, University of Cologne, Cologne, Germany
[2]Alfred Wegener Institute, Helmholtz Centre for Polar and Marine Research, Potsdam, Germany

**Correspondence:** Kerstin Ebell (kerstin.ebell@uni-koeln.de)

**Abstract.**

Accurate precipitation data are essential for understanding the Arctic climate, yet estimates from satellite, re-analysis, or climate models remain uncertain. Ground-based observations, which are sparse in the Arctic, are needed for a better understanding of precipitation processes and, as reference points, can help to characterize uncertainties and improve precipitation estimates. We present extended precipitation measurements at the Arctic research station AWIPEV in Ny-Ålesund, Svalbard, consisting of a Pluvio precipitation gauge, a Parsivel disdrometer, and a micro rain radar. Analyzing four years of data (August 2017–December 2021), we characterized precipitation by amount, type, and frequency and also focused on extreme events. Monthly precipitation at Ny-Ålesund varied widely, from 1 to 155 mm. We also associated the contribution of weather systems, i.e., of atmospheric rivers (ARs), cyclones, and fronts, to precipitation amount. Though ARs (separated or co-located with other weather systems) occur only 8% of the time at Ny-Ålesund, 43% of the total precipitation amount is measured during these events and 22% when only ARs are present. Cyclones contributed 40% (21%) of the total precipitation amount if all (separated) cyclone events are considered. Extreme precipitation events were largely associated with ARs, i.e. in 11 out of 12 cases. Determining precipitation occurrence depends very much on the observation method and the temporal resolution, from 1% (Pluvio at 1-minute resolution) to 21% (micro rain radar) and increased to 38% with daily resolved Pluvio data. Identifying precipitation type solely through Parsivel remains challenging, and a more detailed evaluation using in-situ methods is needed.

## 1 Introduction

Precipitation is a key climate variable that is crucial to the Arctic climate system. It is an integral part of the hydrological cycle and has a direct impact on the Arctic Ocean and land freshwater budget (e.g. Serreze et al., 1995; Cullather et al., 2000; Prowse et al., 2015; Vihma et al., 2016). In the Arctic, most of the precipitation falls as snow (Bintanja and Andry, 2017), altering the surface albedo (Box et al., 2012; Riihelä et al., 2019) and thus the surface energy budget. Snow also directly contributes to the surface mass balance of the cryosphere: e.g., precipitation is the major positive contribution to the mass balance of the Greenland ice sheet (Bring et al., 2016; van den Broeke et al., 2009), as well as to ice caps and glaciers in the Arctic. Snow on sea ice also affects sea ice growth and decay via different snow-sea ice interactions (Serreze and Hurst, 2000).



In the last decades, the Arctic has experienced a rapidly changing climate with a substantial increase in near-surface air
temperature, known as Arctic amplification (Serreze and Francis, 2006; Serreze and Barry, 2011; Wendisch et al., 2023).
A recent study by Zhou et al. (2024) revealed that the Arctic warming between 1979 and 2001 is three times higher than
the global warming. In particular, the Svalbard archipelago is located in the warmest region of the Arctic and reveals the
highest temperature increase (Dahlke and Maturilli, 2017). The potential causes for Arctic amplification are central questions
in Arctic research (Wendisch et al., 2023). In this context, various local feedback mechanisms (e.g., albedo, lapse-rate, water
vapor, Planck, and cloud feedback), as well as remote ones (e.g., oceanic heat and meridional heat and moisture transport) are
discussed (e.g. Goosse et al., 2018; Pithan and Mauritsen, 2014; Wendisch et al., 2023; Mewes and Jacobi, 2019).

The increase in Arctic temperature and the associated mechanisms mentioned before also affect the hydrological cycle of
the Arctic climate system and, thus, precipitation. For example, climate models reveal a substantial increase in precipitation in
the Arctic (McCrystall et al., 2021; Bintanja and Andry, 2017; Bintanja et al., 2020) with rain becoming the most dominant
precipitation type in future (Bintanja, 2018; Bintanja and Andry, 2017). Recent studies have shown that not only the Arctic
mean precipitation will increase in the 21st century but also its interannual variability (Bintanja et al., 2020; Hartmuth et al.,
2023). Thus, extreme precipitation is also becoming more likely. The increase in precipitation is likely caused by different
reasons, i.e., a higher local moisture supply (Bintanja and Selten, 2014; Kopec et al., 2016), increased poleward transport of
atmospheric moisture from lower latitudes (Bengtsson et al., 2011; Bintanja et al., 2020; McCrystall et al., 2021; Pettersen
et al., 2022), but also by a stronger radiative loss of energy to space as shown in a recent study by Pithan and Jung (2021).

Precipitation in Svalbard is highly variable in space and time and strongly influenced by the orography and the large-scale
atmospheric circulation and associated transport routes of air masses (Vikhamar-Schuler et al., 2019; Dobler et al., 2020).
Highest precipitation amounts are typically found at the windward sides of the western mountainous regions (Vikhamar-Schuler
et al., 2019). Vikhamar-Schuler et al. (2019) analyzed downscaled reanalysis data and found that in these regions, annual mean
precipitation is more than 1000 mm per year, while in more sheltered valleys and areas in the central and eastern parts of
Svalbard, yearly precipitation amount can be below 400 mm. These regional differences are also reflected in the precipitation
gauge time series for different stations in Svalbard. At Longyearbyen airport, the yearly mean precipitation amount is lowest
(<200 mm) and about twice as large in Barentsburg and Ny-Ålesund. The different stations show a similar seasonal cycle in
precipitation amount with a minimum in late spring/early summer and a maximum in September/October (Hanssen-Bauer et al.,
2019; Vikhamar-Schuler et al., 2016). For most stations (including Ny-Ålesund), a second maximum can be seen in March. For
all stations in Svalbard, a positive trend in annual precipitation amount has been observed, with significant trends for Bjørnøya,
Hopen and Ny-Ålesund (Hanssen-Bauer et al., 2019). However, since gauge data in Hanssen-Bauer et al. (2019) have not
been corrected for undercatch, trends are also uncertain due to the shift to more liquid precipitation, which is more efficiently
collected by precipitation gauges. This issue has also been raised by Førland and Hanssen-Bauer (2000) and addressed in
more detail by Champagne et al. (2024). Champagne et al. (2024) pointed out that correcting for undercatch is crucial in trend
detection since it significantly impacts the trend magnitude, particularly for snowfall and, thus also, for total precipitation. In
addition to an overall observed increase in precipitation amount, an increase in the frequency of extreme precipitation events has
been found (e.g., Vikhamar-Schuler et al., 2016; Serreze et al., 2015). Based on precipitation gauge data, Serreze et al. (2015),



for example, revealed a significant increase in frequency and intensity for extreme precipitation events at Ny-Ålesund in winter.

Vikhamar-Schuler et al. (2016) further showed that the rate of melt days in winter in Svalbard and the associated precipitation sums have increased. These rain-on-snow events, which have implications for the cryosphere, ecosystem, and infrastructure, have also been studied in further detail (e.g., Hansen et al., 2014, 2019; Peeters et al., 2019; Xie et al., 2024). Such extreme winter events are connected to warm and moist air masses being advected and are also related to cyclones whose number has been found to increase in the last decades (Wickström et al., 2020; Rinke et al., 2017). In particular, atmospheric rivers

(ARs; Ralph et al., 2020) are an essential mechanism for the poleward transport of moisture (Guan and Waliser, 2015). They can significantly impact the Arctic via enhanced precipitation, concurrent heat advection, and increased longwave downward radiation, and subsequent snow and ice melt (e.g., Mattingly et al., 2018, 2020; Bresson et al., 2022). In a recent study by Lauer et al. (2023), the impact of ARs and associated weather systems on Arctic precipitation has been analyzed in detail. Based on ERA5 reanalysis data, precipitation was attributed to ARs, cyclones, and/or fronts for two campaign periods in

early summer 2017 and early spring 2019. Lauer et al. (2023) found that for the early spring campaign, precipitation was dominated by cyclone-related weather systems, while for the early summer period, both ARs and fronts contributed by 40% and 55%, respectively. Furthermore, Dobler et al. (2020) investigated atmospheric circulation types, their future changes, and their impact on precipitation over Svalbard. Based on future climate projections using a regional climate model, they found a distinct increase in precipitation over Svalbard. This increase is not related to changes in circulation type frequencies but rather

due to changes in atmospheric conditions, in particular during cyclonic circulation patterns.

    Even though many studies addressed precipitation in the Arctic and in Svalbard in particular, observing and modeling Arctic precipitation is still very challenging and associated with quite some uncertainties. To gain an Arctic-wide picture of precipitation properties, satellite or model data, i.e., from numerical weather prediction models, reanalyses, or climate simulations, have to be used. However, precipitation is still one of the most uncertain variables in models (Boisvert et al., 2018;

Behrangi et al., 2016). Also, precipitation estimates from satellite measurements are challenging due to uncertainties in retrieval methods, limited observation capabilities close to the surface, and coarse temporal resolution (e.g., von Lerber et al., 2022; Maahn et al., 2014). Continuous, highly temporally resolved ground-based observations of precipitation are thus necessary to better understand precipitation and precipitation-related processes in the Arctic and serve as a reference for satellite and model data. In addition to the classical precipitation gauge observations, ground-based remote sensing has been proven beneficial for

precipitation observations. In particular, measurements of the micro rain radar have been widely used to analyze precipitation in the polar regions (Maahn and Kollias, 2012; Shates et al., 2021; Schoger et al., 2021).

    In this study, we thus present a new comprehensive data set of ground-based precipitation observations at Ny-Ålesund, Svalbard, which includes an OTT Pluvio2L weighing gauge, an OTT Parsivel2 disdrometer and a METEK MRR-2 micro rain radar (MRR). While the Pluvio and the Parsivel provide information on surface precipitation amount and type, the MRR

includes information on the vertical structure of precipitation up to a height of 1 km. These complementary observational records are available in a high temporal resolution, i.e., 1 min, which enables more detailed analyses of precipitation processes, in particular in combination with the further cloud remote sensing instruments at Ny-Ålesund (Chellini et al., 2022). This



data set will thus extend and add to the already existing surface precipitation observations in Svalbard, e.g., the long-term precipitation gauge observations of the Norwegian Meteorological Institute.

In this paper, we will present the results of more than four years of data and focus on some general precipitation characteristics at Ny-Ålesund but also on individual precipitation events. We also link the precipitation amount to weather systems on the synoptic scale, i.e., here, ARs, cyclones, and frontal zones, following the methodology by Lauer et al. (2023). In the next section, the different data sets and methods are introduced. In section 3, four main questions are addressed: How much precipitation falls at Ny-Ålesund, and how is it related to the previously mentioned weather systems? What type of precipi-

tation occurs? How often does it precipitate? Here, we will focus mainly on daily and monthly precipitation statistics. In section 4, individual precipitation events are analyzed in more detail with a focus on extreme events. Conclusions and an outlook are presented in section 5.

## 2   Data and methods

The core instruments used in this analysis are a micro rain radar, a Parsivel, and a Pluvio operated of the University of Cologne

within the Transregional Collaborative Research Centre (TR 172) "Arctic Amplification: Climate Relevant Atmospheric and Surface Processes, and Feedback Mechanisms (AC)[3]" (http://www.ac3-tr.de; Wendisch et al., 2017). All three instruments were installed in 2017 at the German-French AWIPEV research base (78.92308°N, 11.92108°E; 11 m above mean sea level; Fig. 1) that is operated jointly by the Alfred Wegener Institute Helmholtz Centre for Polar and Marine Research (AWI) and the French Polar Institute Paul Emile Victor (IPEV) and is part of the Ny-Ålesund Research Station, Svalbard. In this work,

the data for the years 2017-2021 is considered. A more detailed description of the instruments is given below. We also present auxiliary information included in the analysis.

### 2.1   Pluvio

The Pluvio2L 400 (in the following Pluvio) manufactured by the OTT Hydromet GmbH is an automated weighing gauge with a collecting area of 400 cm². The Pluvio has been installed in the measurement field about 180 m away from the Parsivel and

micro rain radar (Fig. 1). Precipitation falling into the bucket is weighed every 6 s. The difference between the current bucket content and the previously recorded one gives the precipitation amount. The OTT software provides different outputs in a 1 min resolution. In this study, the so-called non-real-time output of the OTT software is used, which is particularly suited for daily and monthly totals (OTT, 2016b). The non-real-time output is delayed by 5 minutes and provides a more precise precipitation sum due to better filtering: fine precipitation is collected over one hour and output after reaching the threshold of 0.05 mm

within that hour. There will be no output if the fine precipitation does not reach the threshold within an hour. The resolution of the precipitation values is 0.01 mm. The measurement uncertainty is the larger value of ±0.1 mm or ±1% (OTT, 2016b). The Pluvio data are available from 2 August 2017 onward (Ebell et al., 2023b). The data availability in each month is generally larger than 90% (Fig. A1a). Months with longer data gaps are March and August 2019, July 2019, October and November 2020, and November 2021. The data gaps are only critical for the monthly precipitation sums of March and July 2019 and




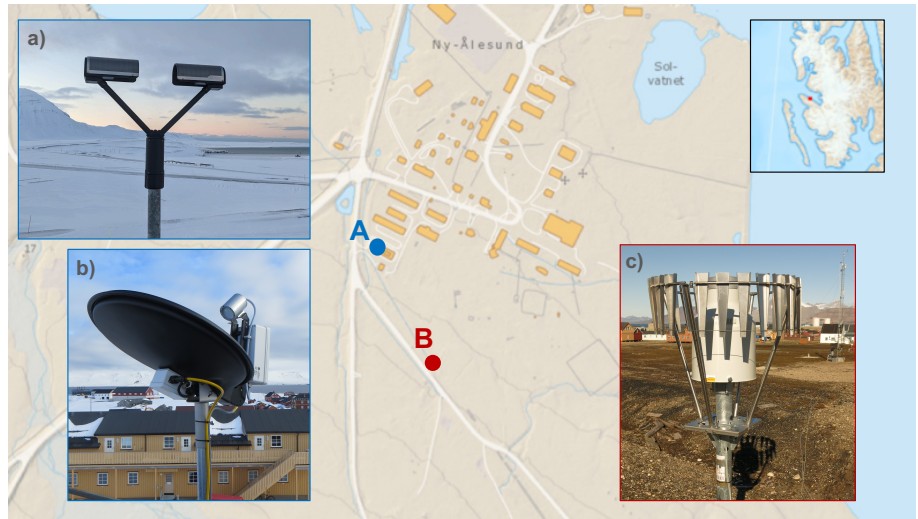

**Figure 1.** a) Parsivel, b) MRR, and c) Pluvio of the University of Cologne at Ny-Ålesund. The Parsivel and MRR are located on the roof platform of the AWIPEV atmospheric observatory (Fig. 1, location A) while Pluvio is installed in the field about 180 m away (Fig. 1, location B). The map of Ny-Ålesund and the map inlet showing the location of Ny-Ålesund in northwestern Svalbard are taken from https://toposvalbard.npolar.no by courtesy of the Norwegian Polar Institute.

October 2020 since significant precipitation has been reported by the micro rain radar and/or the MET Norway precipitation data during the missing periods. Thus, the yearly precipitation sums for 2019 and 2020 are most likely underestimated.

The Pluvio data used in this work were filtered according to the instrument status provided by the OTT software. The software indicates if the instrument runs properly or if an event associated with a "warning" or an "alarm" occurred. All times where the instrument status is associated with an alarm, i.e., an unstable or incorrect weight measurement, have been excluded 130 from the analysis.

Uncertainties in the precipitation amount also arise due to an undercatch of precipitation, particularly solid precipitation and when wind speed is high. Also, blowing snow can affect the measurements. To reduce this uncertainty, the Pluvio is surrounded by a single Alter wind shield, which has been shown to substantially improve the detection of precipitation and reduce the undercatch of precipitation (Nitu et al., 2018): within the WMO Solid Precipitation Intercomparison Experiment 135 (SPICE) project it has been found that overall a shielded gauge improved the catch efficiency by 0.1 to 0.2 compared to an unshielded gauge. We also applied an empirical correction function by Wolff et al. (2015) to the 1-minute precipitation data to further consider wind-induced precipitation losses. This correction function has been developed based on gauge measurements in southern Norway and depends on temperature and wind speed at gauge height (see Eq. 12 in Wolff et al., 2015).





## 2.2 Parsivel

The OTT present weather sensor Parsivel[2] (in the following Parsivel) is an optical laser disdrometer. It provides information on fall speed, size, and type of precipitating particles. The Parsivel consists of two sensor heads with a 30 mm wide, 180 mm long, and 1 mm high laser light strip in-between (OTT, 2016a). The output voltage of the Parsivel is reduced when a precipitation particle falls through the laser beam. The reduction of output voltage is proportional to the particle size. The particle speed is determined by the duration of the voltage signal, i.e., the time the particle needs to enter and leave the laser beam. Measurable

size ranges are between 0.2 and 8 mm for liquid precipitation and between 0.2 and 25 mm for solid precipitation, with 32 size classes in total. Measured fall speeds are in the range of 0.2 and 20 ms$^{-1}$ with 32 particle speed classes. The OTT Parsivel software also retrieves the type of precipitation particles, namely "drizzle", "drizzle with rain", "rain", "rain, drizzle with snow", snow", "snow grains", "soft hail", and "hail". The actual retrieval of the precipitation type is proprietary, but in principle, it relies on the fact that different particle types have different fall speed-size relationships. OTT reports that the differentiation of the

precipitation types drizzle, rain, hail, and snow corresponds to the observations of a weather observer in more than 97% of the cases (OTT, 2016a).

At AWIPEV, the Parsivel has been installed on the western roof platform of the atmospheric observatory (Fig. 1). Data are available since 29 April 2017 (Ebell et al., 2023a). Until May 2021, data coverage is generally quite high (Fig. A1b). From June 2021 onward, longer measurement gaps occurred, and the OTT Parsivel software quality flag often indicated problems with the

glass cover/laser. This was related to humidity condensing inside the instrument. Opening and drying the instrument helped in the short term. Still, the problem re-emerged such that valid measurements are only available to a very limited extent until the end of 2021. Only data for which the quality flag indicated reliable measurements were used. In June 2022, this instrument has been replaced by a new Parsivel.

## 2.3 MRR

The Micro Rain Radar (MRR; type MRR2), manufactured by Meteorologische Messtechnik GmbH (Metek), is a vertically pointing frequency modulated continuous wave (FM-CW) Doppler radar operating at 24.1 GHz (K band). It was installed at Ny-Ålesund at the end of April 2017 next to the Parsivel on the roof platform of the AWIPEV atmospheric observatory (Fig. 1), with data being available since 28 April 2017 (Ebell et al., 2023c). The MRR measures the backscattered signal by falling hydrometeors at 32 range gates. At Ny-Ålesund, the vertical resolution was set to 30 m. Measurements are reported

for the height levels (indicating the lower boundary of the radar bin) from 30 m to 930 m. It should be highlighted that the lowest 2-3 radar bins are affected by near-field effects and also the uppermost radar bin is very noisy. Data at these heights should thus not be used. The temporal resolution of the final data products is 60 s. In the standard data processing of the Metek software, the particle size distribution $N(D)$ is calculated from the spectral reflectivity density with respect to velocity, namely the Doppler spectrum, using an idealized size-fall velocity relation $v(D)$ for rain by Atlas et al. (1973) and Mie theory (Peters

et al., 2002; Maahn and Kollias, 2012). Combining $N(D)$ with $v(D)$ and integration over the drop size distribution results in the rain rate and integration over $N(D)$ in the radar reflectivity factor $Z$. While this method works very well for liquid





precipitation, it is unsuitable for snow (Maahn and Kollias, 2012; Kneifel et al., 2011), which often occurs at Ny-Ålesund. Thus, we used the Improved MRR Processing Tool (IMProToo) by Maahn and Kollias (2012), which includes an improved sensitivity, a correction for aliased data, and reliable values of equivalent radar reflectivity $Z_e$, Doppler velocity, and spectral

width. From this point forward, when mentioning reflectivity, we will be referring specifically to the equivalent radar reflectivity factor $Z_e$. In contrast to the Metek processing, the radar moments are directly calculated from the radar Doppler spectrum. To retrieve a quantitative estimate of precipitation from the MRR, $Z_e$–$S$ or $Z_e$–$R$ relations for snowfall $S$ and rain rate $R$ could be applied. However, this involves prior knowledge of the precipitation phase to decide if $S$ or $R$ must be retrieved. Also the choice of a $Z_e$–$S(R)$ relation introduces further uncertainties in the snow- and rainfall amount. Since this study focuses not on

the evaluation of $S$ and $R$ from MRR (see e.g., Schoger et al., 2021), we only make use of the MRR radar reflectivity as the primary measurement of the instrument for the detection and vertical structure of precipitation. The absolute calibration of the MRR was evaluated with details given in Appendix A. We found that the instrument systematically underestimates reflectivity by 0.6 dB. However, this calibration offset is not applied to the data as it is within the instrument uncertainty. The MRR is very robust in operation (Fig. A1c) with a 100% data coverage in most of the months.

## 2.4 Additional data sets

For the analysis of precipitation type and the correction of precipitation undercatch, we also use the 2 m temperature ($T_{2m}$) and 2 m wind speed measured as part of the Baseline Surface Radiation Network (BSRN) station at Ny-Ålesund (Maturilli, 2020). The data is provided in 1 min resolution. In general, daily mean $T_{2m}$ values are above 0°C in summer and to most extent in September and rarely exceed 10°C (Fig. A2). The lowest temperatures are found in March. This is in line with the long-term

observations at Ny-Ålesund (Dahlke et al., 2020; Maturilli et al., 2013). Fig. A2 also reveals a large variability of daily mean $T_{2m}$ in the cold season with even positive values in winter, indicating the potential for liquid precipitation.

Furthermore, we use the daily precipitation sums provided by the Norwegian Meteorological Institute (MET Norway, https://seklima.met.no) performed manually with an old Norwegian precipitation gauge with a windshield located in the center of the village of Ny-Ålesund and thus about 300 m away from the Pluvio. The measurements are quality controlled but

not corrected for undercatch (Hildegunn D. Nygård, Norwegian Meteorological Institute, pers. comm 11 May 2023). Daily precipitation totals are always valid for 0600 UTC to 0600 UTC the next day, with the time stamp indicating the end of the measurement period.

To associate precipitation to synoptic scale weather events, we analyzed ERA5 reanalysis (Hersbach et al., 2020) data as in Lauer et al. (2023) from 1 August 2017 to 31 December 2021. To this end, ARs, cyclones, and fronts were detected north of

60°N. The details of the weather event detection methods are provided in Lauer et al. (2023), and we give a summary here. The AR detection algorithm applied is the second version of the original method by Guan and Waliser (2015) (Guan et al., 2018). It is based on thresholds in integrated water vapor transport and its geometry. Cyclones are detected based on mean sea level pressure (MSLP) following Sprenger et al. (2017) who use a refined version of Wernli and Schwierz (2006). Finally, frontal systems are calculated from a threshold in the horizontal gradient of equivalent potential temperature (Jenkner et al.,

2010; Schemm et al., 2015). Each reanalysis data grid point (0.25° x 0.25° resolution) is thus classified in terms of the (non-



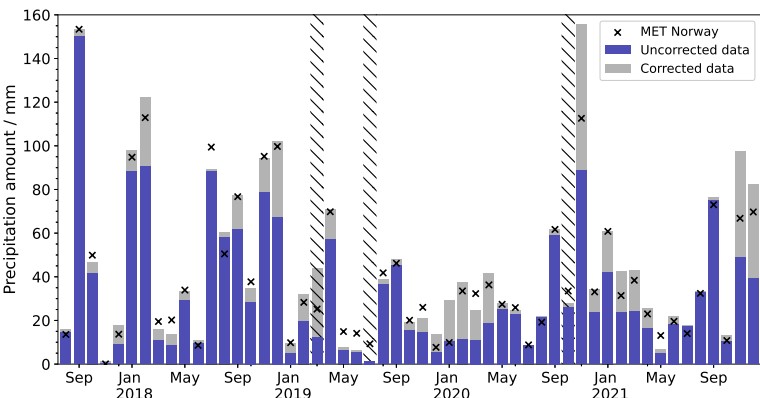

**Figure 2.** Monthly precipitation amount (in mm) from Pluvio based on the uncorrected (blue) and undercatch corrected (see text) data (gray) as well as from MET Norway rain gauge (x symbols). Hatched areas indicate months for which the monthly precipitation sums from Pluvio are underestimated due to measurement gaps.

)occurrence of an AR, cyclone (CY), and/or front (FR). Like the reanalysis data, this weather system classification data set has an hourly temporal resolution. A weather event is then detected for Ny-Ålesund if the grid box in which Ny-Ålesund is located is part of the region of the weather event. The weather systems can occur separately (O-AR, O-FR, O-CY) or simultaneously in different combinations (AR-FR, AR-CY, AR-CY-FR, CY-FR). For one case study (section 4), we also looked at the temporal

development of the different measurements at Ny-Ålesund and also incorporated the integrated water vapor (IWV) from the microwave radiometer HATPRO having a temporal resolution of about 2-3 s (Rose et al., 2005; Nomokonova et al., 2019; Nomokonova et al., 2019).

## 3   General precipitation characteristics

First, we look at the general precipitation characteristics from Pluvio, Parsivel, and MRR, i.e., precipitation amount, frequency,

and type for the period from August 2017 to December 2021, with a focus on monthly statistics. We also relate precipitation amounts to the presence of ARs, CYs, and FRs.

### 3.1   Precipitation amount

Figure 2 depicts the monthly precipitation amount of the uncorrected Pluvio data, the corrected Pluvio data following Wolff et al. (2015), and the MET Norway precipitation sums. Monthly precipitation sums show a large variability ranging from 1 mm

(October 2017) to 155 mm (September 2017). There is no apparent seasonality in precipitation amount from this relatively short period. Considering effects due to undercatch adds 0.5% to 442% to the uncorrected monthly value. In absolute terms, the largest correction is found for November 2020 with an additional 66 mm. The large variability in monthly precipitation sums is





**Table 1.** Annual precipitation amount (in mm) of uncorrected and corrected Pluvio and MET Norway rain gauge data.

|                | 2018 | 2019 | 2020 | 2021 |
|----------------|------|------|------|------|
| Pluvio uncorr. | 618  | 220* | 325* | 330  |
| Pluvio corr.   | 752  | 311* | 495* | 520  |
| MET Norway     | 749  | 313  | 434  | 453  |

\* underestimated due to measurement gaps

also reflected in the large range of yearly precipitation sums (Table 1). 2018 was a very wet year (750 mm). The MET Norway time series since 1975 (not shown) reveals that 2018 was even a record year with the largest annual precipitation amount, while

the long-term annual average of the manual precipitation measurements is 436 mm. In contrast, 2019 was a relatively dry year with a precipitation amount of about 300 mm (Table 1). As mentioned before, the estimates of annual precipitation amounts from Pluvio are likely underestimated for 2019 and 2020 due to measurement gaps during some precipitation periods. However, the MET Norway data also indicate a relatively low annual precipitation amount for 2019. The values of monthly precipitation amount derived from Pluvio agree reasonably well with the manual MET Norway observations. However, a direct comparison

is difficult due to the different measurement systems, locations, and thus, different wind conditions having a different impact on precipitation undercatch. For some months, the (uncorrected) MET Norway measurements show even higher values than the corrected ones from Pluvio. A detailed comparison of the instruments would also require a precipitation correction of the MET Norway time series, which is not the scope of this paper. However, Jacobi et al. (2019) have compared the manual MET Norway precipitation observations, the Pluvio measurements as well as automatic precipitation measurements of a Geonor T-

200 weighing gauge (also operated by MET Norway) for a full hydrological year (Sep 2017–Sep 2018) and also took different correction methods into account. The Geonor is located in the same field as the Pluvio, about 100 m apart. An excellent agreement between Pluvio and Geonor annual precipitation sums had been found, while the manual precipitation observations revealed much higher values for both uncorrected and corrected data. The same tendency can also be seen in the present study, which underlines the challenge of precipitation correction and its uncertainty.

Moving to shorter temporal scales, we also analyzed daily precipitation sums of the corrected Pluvio data (Fig. 3). 50% of the daily precipitation sums have values lower than 1.3 mm (gray line in Fig. 3) and contribute only about 5% to the total precipitation at Ny-Ålesund (black line in Fig. 3). Very small precipitation amounts or trace precipitation, i.e., small but immeasurable daily precipitation events, are still challenging for observations and models. Boisvert et al. (2018), who defined trace precipitation as days with less than 1 mm precipitation, showed large differences in the occurrence and annual

amount of trace precipitation over the Arctic Ocean between eight reanalyses. At Ny-Ålesund, trace precipitation (i.e., non-zero daily precipitation amount < 1 mm) is reported from the corrected Pluvio data for about 44% of the analyzed period. Trace precipitation is thus a common feature of the atmospheric state at Ny-Ålesund. The annual trace precipitation amounts for the years 2018-2021 are between 20 to 30 mm. Compared to the annual precipitation amount, these values are rather small. E.g. for 2021, the annual trace precipitation amount is 5.5% of the total precipitation amount at Ny-Ålesund. In particular, in other



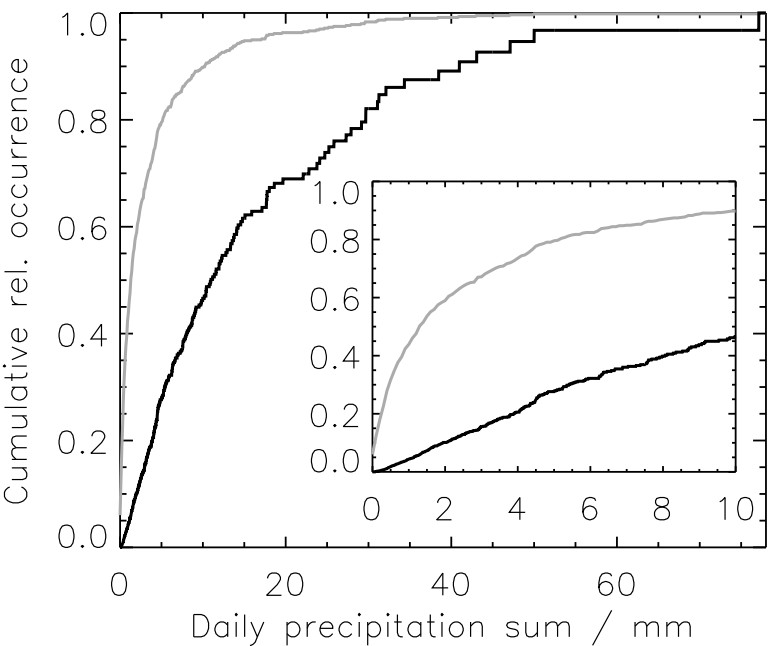

**Figure 3.** Cumulative relative occurrence of daily precipitation sums (grey) and cumulative relative contribution of daily precipitation sums to total precipitation amount (black) based on the corrected Pluvio data for the time period 1 August 2017 to 31 December 2021. The inlet is a zoom-in for daily precipitation sums below 10 mm.

dry Arctic regions, trace precipitation can make up a substantial proportion of the total precipitation amount (Boisvert et al., 2018). Trace precipitation could be associated with the frequent occurrence of low-level mixed-phase clouds in conjunction with katabatic winds (Gierens et al., 2020), with the dry katabatic flow leading to the sublimation of a large portion of the precipitating mass. Only 5% of the days with precipitation measured by Pluvio have a daily precipitation amount larger than 13.3 mm. These 5% of precipitation days contribute about 45% to the total precipitation at Ny-Ålesund.

As discussed earlier, large-scale weather patterns substantially impact Arctic precipitation. We thus related the measured precipitation amount by Pluvio with the presence of ARs, cyclones, and fronts. Fig. 4 depicts the monthly occurrence of these weather systems based on the methodology by Lauer et al. (2023) and the corrected precipitation amount from Pluvio associated with these. ARs (separated and co-located) typically occur less than 15% of the time in a month. A higher occurrence has been found for Sep 2017 (49%) and July 2018 (37%). Front occurrence (separated and co-located) shows yearly maxima of more

than 20% in summer or late summer, which might be related to the differential heating of the Arctic Ocean and the snow-free land as well as coastal orography which supports baroclinicity (Serreze and Barry, 2014). Separated and co-located cyclones occur all year long with a mean value of 20%. When looking at the contribution of these weather systems to the monthly precipitation amount (Fig. 4b,c), we find that the largest contributions are from the AR and cyclone classes. In particular, in





**Table 2.** Contribution of atmospheric rivers (AR), cyclones (CY), and fronts (FR) to the precipitation amount (in %) of the different years and of the whole time period considered (1 August 2017 - 31 December 2021). Weather systems can be separated (only, O) or co-located.

|  | 2018 | 2019 | 2020 | 2021 | whole time |
|---|---|---|---|---|---|
| O-AR | 29 | 24 | 11 | 15 | 22 |
| AR-FR | 7 | 2 | 6 | 3 | 6 |
| AR-CA | 8 | 5 | 9 | 5 | 8 |
| AR-CY-FR | 5 | 9 | 8 | 2 | 6 |
| O-CY | 18 | 26 | 26 | 24 | 21 |
| CY-FR | 3 | 3 | 4 | 6 | 4 |
| O-FR | 3 | 4 | 5 | 5 | 4 |
| residual | 27 | 27 | 30 | 40 | 29 |

the very wet month Sep 2017, almost all precipitation, i.e., 145 mm, can be related to the (co-)occurrence of ARs. For the
month with the highest precipitation amount, i.e., November 2020, both AR and cyclone classes contribute together to about
80% of the total precipitation amount, with the only-cyclone class even dominating. The monthly precipitation amount from
fronts (separated and co-located) is, on average, less than 9 mm and shows a distinct contribution to monthly precipitation
in only a few months, e.g., August 2020 (O-FR: 35%). Quite some precipitation cannot be attributed to any of these weather
patterns with a mean monthly value of 33%. This residual is generally larger from early autumn to early spring. The annual
and whole-time statistics (Table 2) show that separated ARs (O-AR) and cyclones (O-CY) each contribute about 20% to the
total precipitation with quite some variability from year to year. E.g., in 2018, O-ARs (co-located ARs) contributed almost
30% (49%). Fronts only seem to play a minor role in the precipitation amount at Ny-Ålesund: separated fronts contribute only
about 4% to the total precipitation. Only in combination with ARs and cyclones the value increases to 20%.

## 3.2  Precipitation type

Information on surface precipitation type is provided by the Parsivel. In this study, the Parsivel types "snow", "snow grains",
"soft hail" and "hail" are summarized in a "solid" precipitation class, and the types "drizzle", "drizzle with rain", and "rain" in
the "liquid" precipitation class. "Rain, drizzle with snow" is the only Parsivel class that is of "mixed"-phase precipitation. In
fact, this "mixed"-phase class occurs very rarely with a mean monthly occurrence of 0.12% only and a maximum occurrence
of 1.5% in Sep 2017. Before discussing the precipitation type occurrence in more detail, we will first have a closer look at the
Parsivel performance. Since no reference data for precipitation type is available, we have combined the Parsivel with the 2 m
temperature measurements to check for consistency of the retrieved precipitation type and plotted the occurrence of liquid,
solid, and mixed precipitation from Parsivel data as a function of 2 m temperature $T_{2m}$ using the 1 min resolved data (Fig. 5).
This makes sense since temperature is often used as a proxy for the discrimination of solid and liquid precipitation (Champagne
et al., 2024; Kneifel et al., 2022). Fig. 5 shows that the transition between solid to liquid precipitation occurs between 0°C to



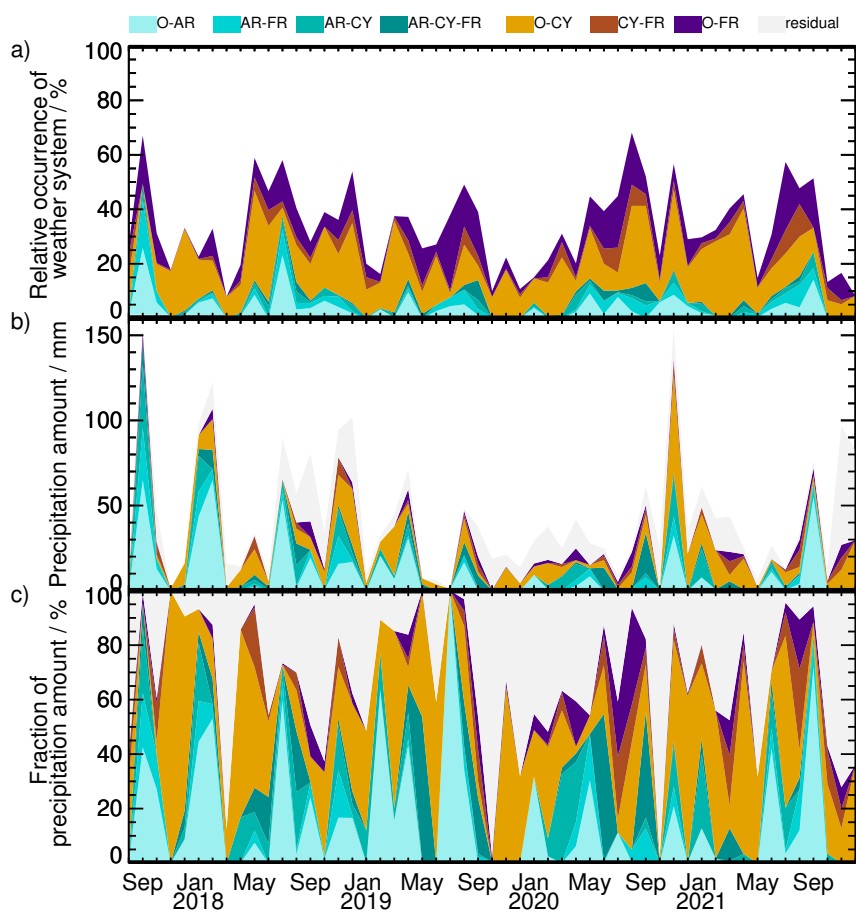

**Figure 4.** a) Relative monthly occurrence (in %) of weather systems at Ny-Ålesund. Atmospheric rivers (AR), cyclones (CY), or fronts (FR) can be separated (only O) or co-located. Monthly precipitation amount associated with different weather types in b) absolute and c) relative values with respect to the total monthly precipitation amount. Precipitation, which is not associated with any of these weather types, is denoted as "residual". Note that the different classes are stacked in the plots.

3°C with an equal occurrence of solid and liquid precipitation around 2°C. When using all Parsivel data (dotted lines in Fig. 5), liquid precipitation is detected by Parsivel even for temperatures far below 0°C and solid precipitation even for temperatures larger than 5°C. Wind and turbulence can affect the particle velocity when passing through the Parsivel laser beam such that the measured velocity does not correspond to the true fall speed of the precipitation particles. Subsequently, this effect will result in a misclassification of the measured particles. Filtering the data by removing cases with 2m wind speeds larger than



5 ms$^{-1}$ (solid lines in Fig. 5) results in a smoother transition from solid to liquid precipitation removing liquid occurrence at very low temperatures and almost all solid precipitation at temperatures larger than 3°C. Even after filtering, the Parsivel data shows an unexpectedly higher liquid occurrence around -3 to -2°C. Looking at these cases in more detail reveals that all these situations are during periods when solid precipitation has been detected by Parsivel as well (not shown). Detected particle sizes during these cold "liquid" events are relatively small, with a mean volume equivalent diameter of 1.3 mm only. A possible

temperature inversion resulting in positive temperatures in upper height levels could be excluded from radiosonde profiles. We also checked similar cases for more recent dates for which measurements by a video-in situ snowfall sensor (VISSS; Maahn et al., 2024) at Ny-Ålesund are available. The VISSS has been installed at Ny-Ålesund in September 2021 and is operated in the measurement field about 140 m northwest of Pluvio. Visual inspection of the pictures of the particles taken by VISSS for a case on 5 May 2023 clearly showed that only solid precipitation was present (Max Maahn, University of Leipzig, pers.

comm 25 August 2023). We thus assume that the Parsivel algorithm falsely classifies the signal as "rain" or "drizzle" for this temperature regime. Interestingly, in this temperature regime, Chellini et al. (2022, 2023) found that low-level mixed-phase clouds at Ny-Ålesund produce small fast-falling ice particles, which could potentially be misinterpreted as drizzle by Parsivel.

     According to the discussion of the previous paragraph, we refined the precipitation classification for the following analyses: precipitation is assumed to be purely solid for $T_{2m} < $ -2°C and purely liquid for $T_{2m} > $ 4°C. For -2°C $\leq T_{2m} \leq$ 4°C, we use

the Parsivel information even though we can not rule out completely wind effects and uncertainties due to the OTT classification algorithm. In Fig. 6, the fraction of monthly precipitation occurrence being solid, liquid and mixed-phase with respect to all cases with a precipitation signal from Parsivel is shown. Typically, solid precipitation dominates from October to May and liquid precipitation from June to September, with May/June and October/November being the transition months. January and November 2018 reveal an exceptionally high fraction of liquid precipitation occurrence of 50% which is connected to positive

temperatures on a few days in these months.

     In section 3.1, the total monthly and yearly precipitation amount has been discussed, but the question remains: how much of the precipitation amount can be attributed to solid, liquid, or mixed precipitation? We thus combined the Pluvio, Parsivel, and temperature information to allocate the precipitation amount to a precipitation phase. Also, here, precipitation is assumed to be purely solid for $T_{2m} < $ -2°C and purely liquid for $T_{2m} > $ 4°C. For -2°C $\leq T_{2m} \leq$ 4°C, we check the Parsivel classification

within a time window of ±10 min. This is reasonable since the instruments are not located directly next to each other. If only solid (liquid) precipitation has been detected within this time interval, the precipitation amount is assumed to be solid (liquid). If both phases have been detected by Parsivel, the precipitation amount is attributed to a "mixed-phase" type. If the Parsivel does not detect any precipitation or no Parsivel measurements are available, the precipitation phase is "unknown". Allocation of precipitation amount to a certain precipitation phase is actually quite challenging at Ny-Ålesund because a substantial amount

of precipitation occurs in the transition regime between -2°C and 4°C, i.e. about 47% of the total precipitation amount during August 2017 and December 2021. Fig. 6 (filled contours) shows the fraction of monthly precipitation amount in each phase. Generally, the precipitation amount of each phase class follows the monthly fraction of precipitation type occurrence. Since the mixed-phase precipitation type, as detected by Parsivel, only seldom occurs, most of the mixed-phase precipitation amount is because both solid and liquid precipitation occur within the ±10 min time window considered. Since the data coverage of



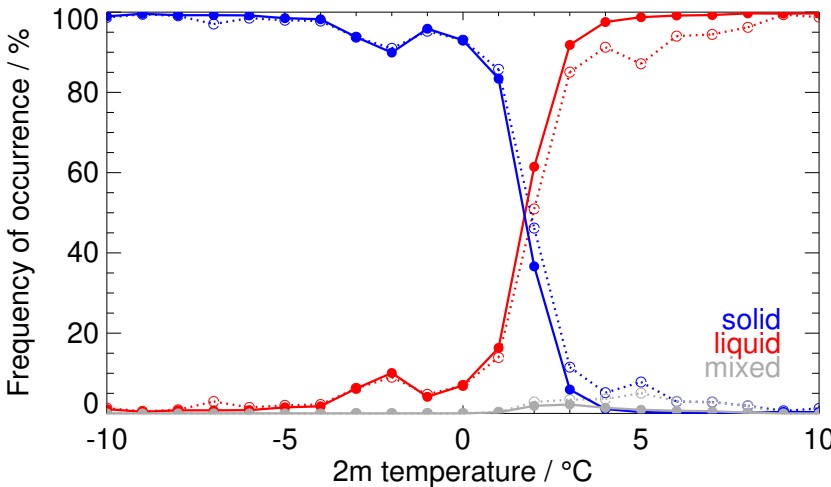

**Figure 5.** Frequency of occurrence of liquid (red), solid (blue), and mixed (gray) precipitation derived from Parsivel as a function of 2 m temperature. Results are shown for all data (dotted line with circles) and data when 2m wind speed $<5$ ms$^{-1}$ (solid line with filled circles). The values have been normalized with the total number of observations within each temperature class. The temperature bin size is 1°C.

Parsivel is very poor from June 2021 onward (see Section 2.2), much of the measured precipitation amount by Pluvio can not be allocated a precipitation type, resulting in higher numbers of the "unknown" class. More than 50% of the monthly precipitation amount is purely solid from October to April except for October 2017, January and November 2018, and April 2019. Only for a few months during late summer/early autumn, the monthly precipitation amount is almost entirely liquid. Nevertheless, liquid and mixed-phase precipitation can also dominate precipitation amounts in other months, e.g., in January 2018 with nearly 90%

and in November 2018 and May 2019 with about 60%. When looking at the annual precipitation amount, about 22% to 30% is found to be purely liquid, with another 4 to 27% being of mixed-phase type (Table 3).

     The previously mentioned rain-on-snow (ROS) events are of particular interest since they can have severe implications for wildlife and Arctic communities. We investigated ROS events using daily precipitation sums during the cold season months November to March. A ROS event is defined as a day with the sum of liquid and mixed-phase daily precipitation exceeding

1 mm. The partitioning into liquid and mixed-phase precipitation follows the same strategy as the monthly sums. Except for the relatively cold 2019/2020 winter (Fig. A2), for which no ROS event has been found, 5-6 ROS events have been detected for each Nov-Dec period, i.e., 16 in total for the whole analysis period.

### 3.3 Precipitation frequency

     From the previous sections, it became clear that determining precipitation amount and type is challenging. However, even

the detection of precipitation and, thus, the determining precipitation frequency are subject to great uncertainty and are very sensitive to the method applied. This can be seen in the monthly precipitation frequency determined by Parsivel, Pluvio, and





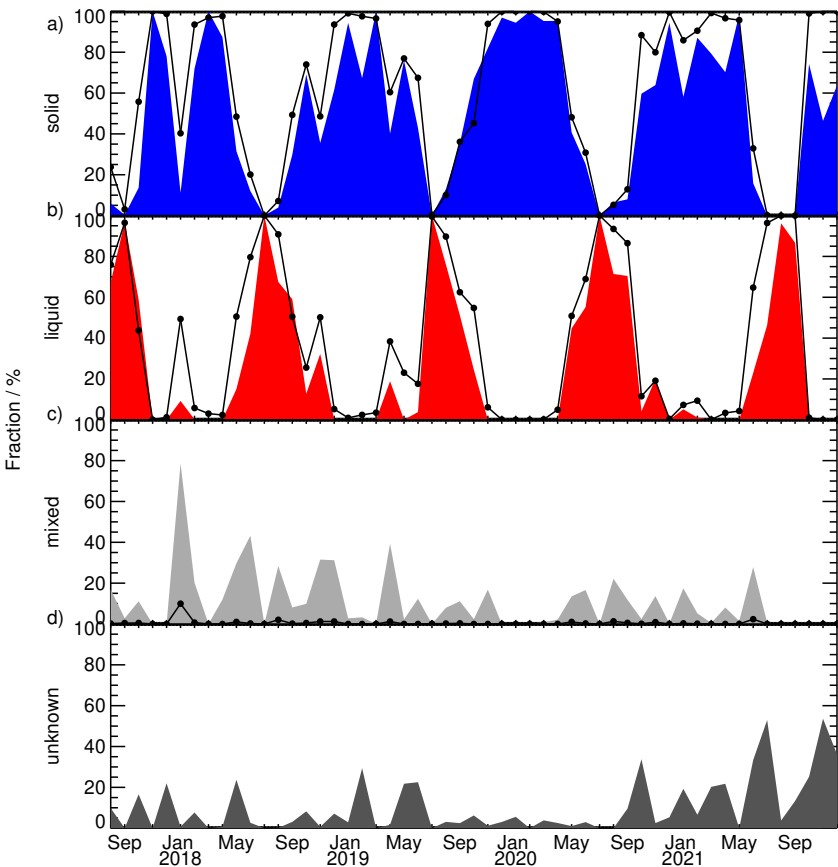

**Figure 6.** Fraction of monthly precipitation occurrence (black lines/dots) and monthly precipitation amount (filled contours) being of a) solid, b) liquid, c) mixed, and d) unknown type. The precipitation type was determined using Parsivel and 2 m temperature (see text for more details). The fraction of monthly precipitation occurrence is given with respect to all cases with a precipitation signal from Parsivel. In contrast, the fraction of monthly precipitation amount is given with respect to all cases with a precipitation signal from Pluvio.

MRR (Fig. 7). Monthly precipitation frequency has been determined from the times when a signal has been recorded by the corresponding instruments relative to the total number of measurements. To this end, we used all 1 min resolved data within a month. Since precipitation amount is often available on a daily basis only, we also calculated the monthly precipitation
frequency based on daily Pluvio values with a precipitation amount >0 mm and >1 mm. For the MRR, we took the measured radar reflectivity $Z_e$ at the lowest height bin with valid measurements, i.e., at 120 m. We counted all cases with $Z_e >$ -10 dBZ which corresponds to the sensitivity of the instrument at that height (Fig. 8). To account for the sensitivity of the results to the $Z_e$ threshold applied, the corresponding values for a more conservative threshold of -5 dBZ are also shown. To understand which rain or snowfall rate a value of -5 or -10 dBZ actually represents, $Z_e - R$ and $Z_e - S$ relationships must be applied.
If we exemplarily use the $Z_e - R$ relation by Tokay et al. (2009) ($Z_e = 129R^{1.5}$), -10 dBZ (-5 dBZ) corresponds to a rain



**Table 3.** Annual precipitation amount (in mm) of Pluvio and separated into precipitation phase. Numbers in parentheses are the percentages relative to the total annual precipitation (in %). See text for more details.

|  | 2018 | | 2019 | | 2020 | | 2021 | |
|---|---|---|---|---|---|---|---|---|
| Pluvio corr. | 752 | | 311* | | 495* | | 520 | |
| liquid | 229 | (30) | 75 | (24) | 127 | (26) | 114 | (22) |
| solid | 285 | (38) | 174 | (56) | 300 | (61) | 243 | (47) |
| mixed | 206 | (27) | 42 | (14) | 43 | (9) | 23 | (4) |
| unknown | 32 | (4) | 18 | (6) | 25 | (5) | 141 | (27) |

\* annual sum underestimated due to measurement gaps

rate of about $0.01\,\mathrm{mm\,h^{-1}}$ ($0.02\,\mathrm{mm\,h^{-1}}$). If we use for example the three bullet rosette $Z_e - S$ relationship by Kulie and Bennartz (2009) ($Z_e = 24.04 S^{1.51}$), which was also used in the study by Maahn et al. (2014), corresponding snowfall rates are $0.03\,\mathrm{mm\,h^{-1}}$ and $0.06\,\mathrm{mm\,h^{-1}}$, respectively. In principle, only a few mm sized snowfall particles (aggregates) per cubic meter are sufficient to be detected by the MRR.

The monthly precipitation frequency from the MRR ranges from 1% to 38%. Averaging over the whole measurement period results in a mean value of 21%. From the short time series, no seasonality can be deduced. The more conservative threshold of -5 dBZ slightly reduces the monthly values, typically about 2 to 4 percentage points, and the whole time mean precipitation frequency to 18%.

     Since the Parsivel is also sensitive to single precipitation particles, the derived precipitation frequency is also high, with
maximum values of about 18%. However, these numbers are generally lower than those from MRR, which also has a much larger sampling volume: e.g., with a beam width of $1.5°$, the observed volume between 120 m and 150 m, for example, is about $300\,\mathrm{m^3}$. Also, the time series of monthly precipitation frequency differs between Parsivel and MRR, resulting in a correlation coefficient (May 2017 - May 2021) of 0.72 only.

     Surface precipitation, i.e., any signal in the Pluvio measurements, is recorded only in very few cases. Using the 1 min
resolved Pluvio time series to determine precipitation frequency results in monthly values of up to 5% only. Using daily resolved Pluvio data increases the monthly precipitation frequency to 4%–63% and, if a threshold of 1 mm is applied, to 4%–46%

     Considering the months August 2017 to May 2021, for which all instruments provide sufficiently large data coverage, the mean precipitation frequency is 8% (Parsivel), 1% (Pluvio, 1 min resolution), 38% (Pluvio, daily resolution, >0 mm), 22%
(Pluvio, daily resolution, >1 mm), and 21% (MRR with -10 dBZ threshold), respectively.

     We want to emphasize that we do not evaluate the derived precipitation frequency of each method. Depending on the individual scientific question, the precipitation frequency by one or the other method might be more relevant. For example, for hydrological applications, the precipitation occurrence based on Pluvio measurements might be of more interest. Still, precipitation occurrence as detected by the MRR is relevant for analyzing the microphysical processes of clouds and precipitation.





In contrast to Parsivel and Pluvio, the MRR also provides information on vertical precipitation structure. Fig. 8a depicts the 2D histogram of MRR $Z_e$ for the whole analysis period. Here, we plotted the data of all height levels included in the IMProToo data set. From the 2D histogram, it becomes clear that measurements are unreliable below 120 m and above 900 m and should not be used for these levels. Looking at the other height levels, we find a median $Z_e$ profile with a vertically constant value of about 6 dBZ. The same feature has also been found for MRR measurements taken at the Sverdrup Research Station at Ny-Ålesund from March 2010 to March 2011 (Maahn et al., 2014) as well as for X band radar observations at Iqaluit in the Canadian Arctic (Henson et al., 2011). When looking more closely at the signal frequency for different $Z_e$ thresholds (Fig. 8b), we can see that, in particular, higher $Z_e$ values are equally distributed in the vertical. For lower $Z_e$ thresholds, a maximum in signal frequency is found for a height of about 540 m. The decrease of signal frequency below, i.e., about 2 percentage points, is likely related to sublimation effects. By simply counting cases where no MRR signal has been reported at 120 m but in any other height layer above ($Z_e$ >-10 dBZ), we find a mean "sublimation" occurrence of 7%. This number should be taken as an upper estimate since we do not consider advection and wind effects, as well as tilted fall streaks.

## 4  Event-based precipitation analysis and extreme events

For the synergetic analysis of all three instruments, it is useful to consider individual precipitation events. In the last part of this study, we would like to address the following questions: what kind of precipitation events contribute most to the precipitation amount at Ny-Ålesund, what is the duration of precipitation events, and which role do extreme precipitation events play in this respect? As a first step, "precipitation event" has to be defined. Here, we take the following approach: based on the 1 min resolved data, we first check when MRR $Z_e$ at 120 m height is greater than -10 dBZ. Periods with $Z_e$ greater than -10 dBZ need to be more than 30 min apart to be counted as two individual MRR events. In this way, we allow for short interruptions in precipitation since precipitation might still be associated with the same larger-scale cloud/weather system. The threshold of 30 min is somewhat arbitrary, and the sensitivity of the results to this threshold will be discussed later. A further criterion for a precipitation event is that Pluvio has detected precipitation during the MRR event. Here, we extended the considered period by ±10 min due to the spatial distance of the Pluvio and MRR measurements. While 3081 MRR events were detected between 1 August 2017 and 31 December 2021, only 645 of them (∼21%) contained precipitation measured by Pluvio. For 162 MRR events (∼5.2%), no information on precipitation amount is available. The 645 precipitation events will be analyzed in more detail in the following.

As seen from Fig. 7, the MRR is quite sensitive for detecting precipitating particles. The question, therefore, arises for which $Z_e$ values precipitation is actually detected on the ground. For each event, we thus calculated the maximum and median $Z_e$ at 120 m height. Figure 9 depicts the relative occurrence of maximum and median $Z_e$ for all precipitation events and for all MRR events (without the surface precipitation criterion applied). We find that maximum Ze values during a precipitation event are typically between 10 to 40 dBZ, and median values centered around 5 dBZ. These results are independent of the temporal separation threshold of 30 min, and the same numbers are found for thresholds of 15 min and 60 min (not shown).



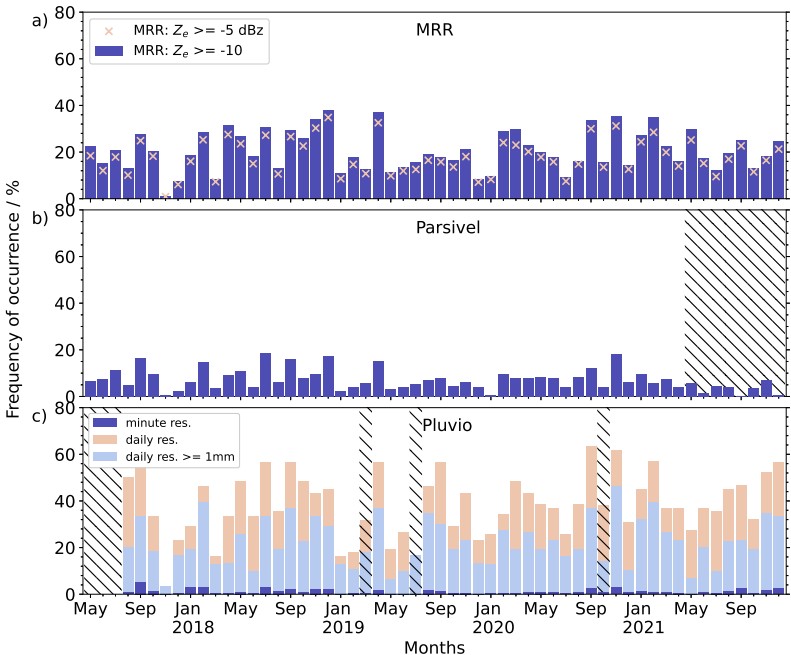

**Figure 7.** Monthly frequency of precipitation occurrence for a) MRR, b) Parsivel, and c) Pluvio. For the MRR, precipitation is assumed to occur if $Z_e$ at 120 m height is larger than -10 dBZ (-5 dBZ; x symbols). The monthly values have been calculated from the 1 min resolved data (dark blue in all panels). For Pluvio, monthly precipitation occurrence has also been calculated based on daily precipitation amounts >0 mm (light red) and >1 mm (light blue). Hatched areas indicate months when the monthly values are unknown/unreliable due to missing measurements.

When looking at the cumulative relative occurrence of event duration (Fig. 10a), one can see that, as expected, the distribution of event duration is sensitive to the temporal separation threshold applied: using a 15 min threshold results in a higher occurrence of shorter events. In contrast, a 60 min threshold shifts the distribution to more extended events. Depending on the

event separation criterion, 50% of all precipitation events have a duration of less than 4.2 h to 9.6 h and maximally contribute to the total precipitation amount with 11%. Events shorter than one hour rarely occur, i.e. less than 6% of all events, and have a negligible contribution to the total precipitation (<1%). Events covering more than one day represent only about 5% to 15% of all events but contribute to the total precipitation by about 37% to 60%.

If we analyze the events in terms of their precipitation amount (Fig. 10b), we can see that the cumulative distribution is less

sensitive to the event separation criterion. About 54% (85%) of all precipitation events have a precipitation amount of less than 1 mm (5 mm). They contribute to about 6% (27%) of the total precipitation amount. Events with a precipitation amount of more than 20 mm rarely occur, i.e., only 3% of all events but make up 41% of the total precipitation amount. Focusing on




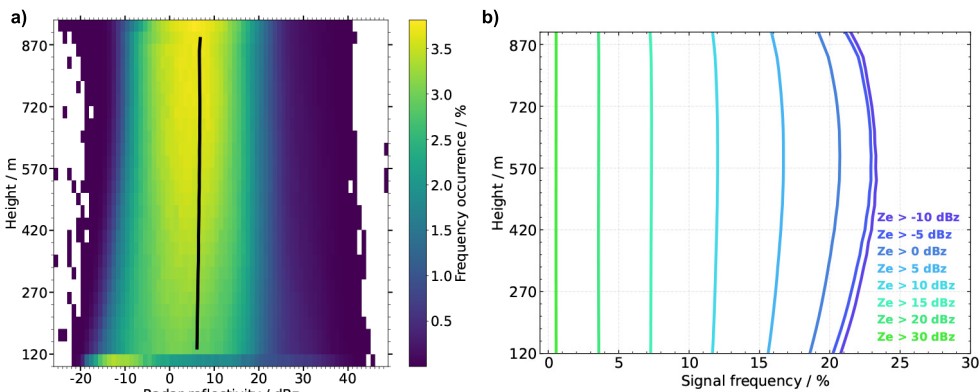

**Figure 8.** a) 2D histogram of MRR $Z_e$ as a function of height for May 2017 to December 2021. The occurrence has been normalized by the total number of MRR profiles. The black solid line indicates the median $Z_e$ profile. b) Signal frequency of each height layer for May 2017 to December 2021. Each line represents the signal frequency with different dBz thresholds ranging from -10 to 30 dBz.

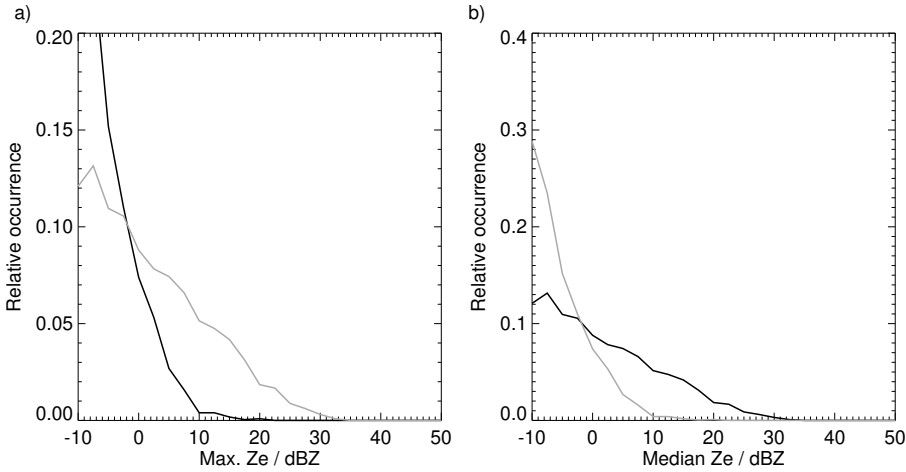

**Figure 9.** Relative occurrence of a) maximum and b) median $Z_e$ for all MRR precipitation events from 1 August 2017 to 31 December 2021 with (black) and without (gray) precipitation measured by Pluvio.

extreme events, i.e., here the upper 2% of events with the highest precipitation amounts (12 events in total), we find that all these events exceed 31.7 mm and contribute 28% to the total precipitation amount of all events (Table 4). In terms of duration,

they vary between 25 and 67 h. Interestingly, almost all events (except for one) are associated with the occurrence of an AR. However, also cyclones and fronts are typical accompanying features for these extreme events.

The high temporal resolution of the precipitation observations allows for a more detailed analysis of the temporal development of these extreme events. While the thorough analysis of all extreme precipitation cases listed in Table 4 is beyond the scope of this paper and will be addressed in a follow-up study, we will exemplarily have a closer look at event #1 having the





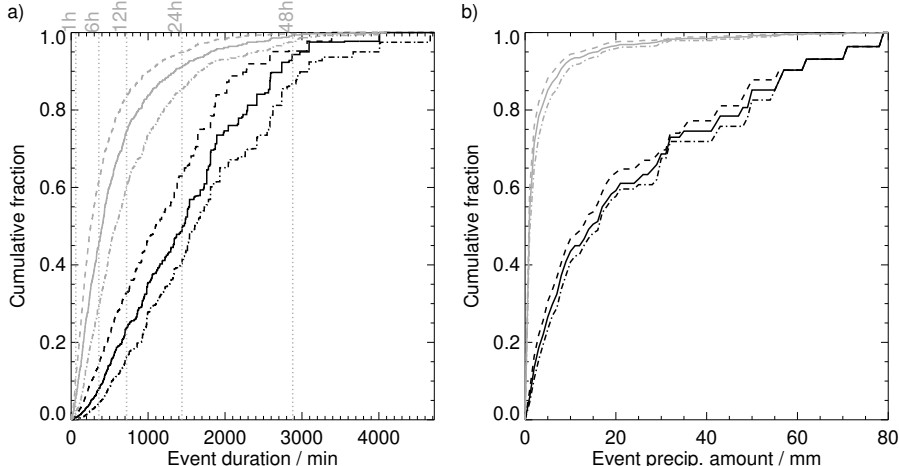

**Figure 10.** Cumulative relative occurrence of all precipitation events from 1 August 2017 to 31 December 2021 (gray) and cumulative relative contribution of precipitation events to total precipitation amount (black) as a function of a) event duration (in min) and b) event precipitation amount (in mm). Results for precipitation events with a 15 min (60 min) separation threshold are indicated with dashed (dashed-dotted) lines.

**Table 4.** Upper 2% of events with the highest precipitation amount ranked in terms of their precipitation amount. In addition to the precipitation amount and duration of the precipitation event, the weather systems detected at least once during the event are reported, i.e., atmospheric river (AR), cyclone (CY), and front (FR).

| # | Period (Date/Time UTC) | Amount (mm) | Duration (h) | Detected weather system | | |
|---|---|---|---|---|---|---|
| 1 | 12 Jan 2018, 20:37 - 14 Jan 2018, 02:00 | 79 | 29 | AR | CY | FR |
| 2 | 26 Feb 2018, 10:25 - 27 Feb 2018, 16:37 | 71 | 30 | AR | – | FR |
| 3 | 09 Nov 2021, 00:56 - 10 Nov 2021, 17:09 | 61 | 40 | – | CY | – |
| 4 | 01 Sep 2017, 19:58 - 03 Sep 2017, 17:35 | 57 | 46 | AR | CY | FR |
| 5 | 18 Dec 2018, 01:31 - 20 Dec 2018, 05:02 | 55 | 52 | AR | CY | FR |
| 6 | 17 Nov 2018, 19:21 - 19 Nov 2018, 14:24 | 50 | 43 | AR | CY | FR |
| 7 | 14 Nov 2020, 05:36 - 17 Nov 2020, 00:25 | 50 | 67 | AR | CY | FR |
| 8 | 27 Nov 2020, 18:02 - 29 Nov 2020, 01:34 | 48 | 32 | AR | CY | – |
| 9 | 18 Sep 2021, 12:36 - 19 Sep 2021, 13:47 | 43 | 25 | AR | – | FR |
| 10 | 30 Aug 2018, 06:08 - 31 Aug 2018, 07:46 | 42 | 26 | AR | CY | FR |
| 11 | 09 Jul 2018, 14:31 - 11 Jul 2018, 02:45 | 34 | 36 | AR | – | FR |
| 12 | 24 Sep 2017, 20:47 - 26 Sep 2017, 00:44 | 32 | 28 | AR | – | FR |

maximum precipitation amount of 79 mm within 29 h. Figure 11 depicts the time series of the different precipitation observations, the measurements of 2 m temperature, wind speed, and IWV, as well as the occurrence of weather systems detected



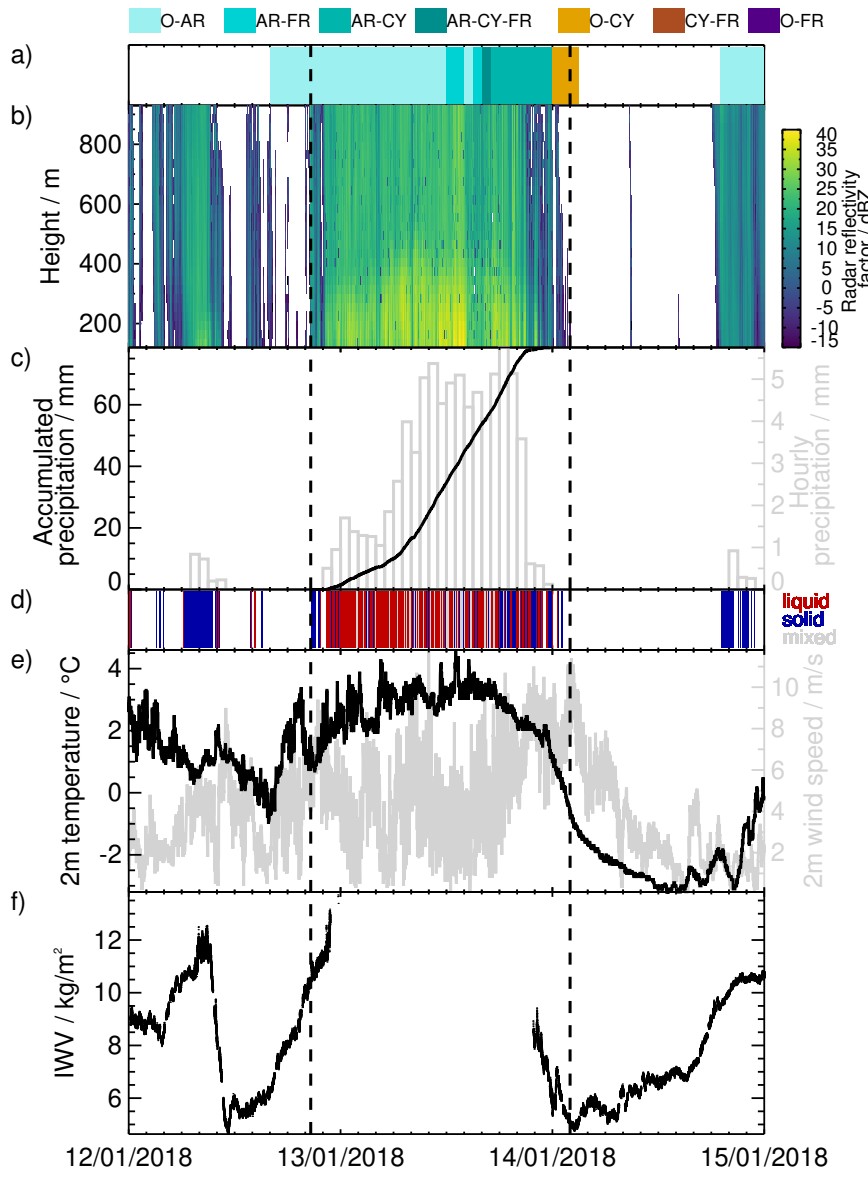

**Figure 11.** Extreme precipitation event on 12 - 14 January 2018. a) Occurrence of weather systems as described in Fig. 4, b) MRR radar reflectivity (in dBZ), c) hourly accumulated precipitation (grey bars) and accumulated precipitation during the event (black line) (in mm), d) Parsivel precipitation type, e) 2 m temperature (black line, in °C) and 2m wind speed (grey line, in m/s), and f) integrated water vapor (IWV) from HATPRO (in kg/m$^2$). The vertical dashed lines indicate the start and end time of the precipitation event.

from ERA5. The precipitation event was accompanied by a substantial increase in water vapor on 12 January 2018 from about 5 kg/m$^2$ to more than 12 kg/m$^2$. With the onset of the precipitation at the end of that day, the HATPRO microwave radiometer



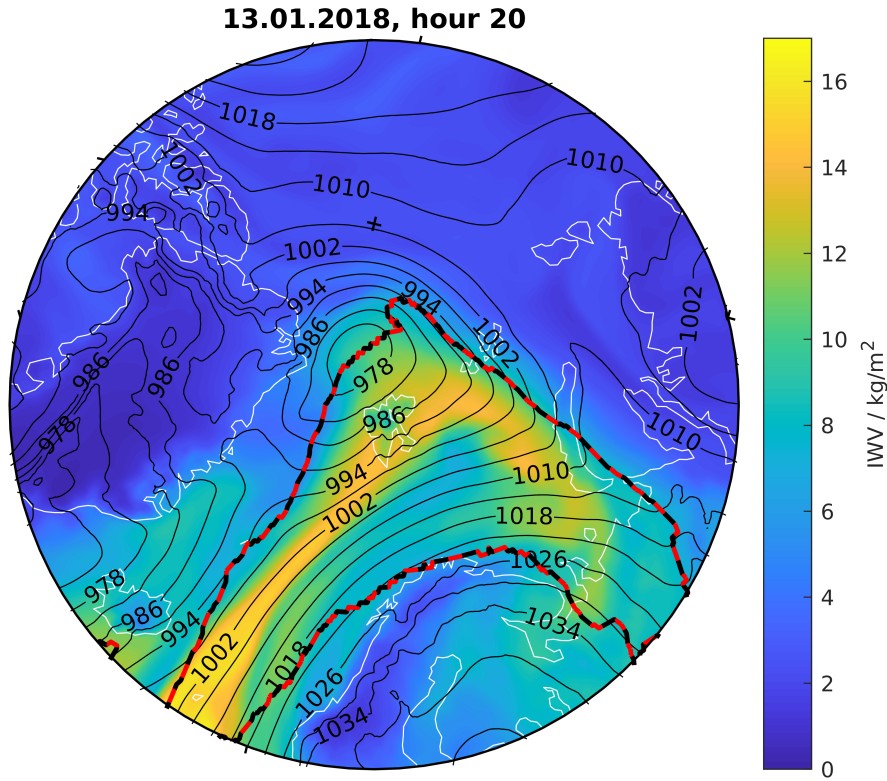

**Figure 12.** IWV (colors) and MSLP (black contour lines) on 13 Jaunuary 2018 at 20 UTC from the ERA 5 reanalysis. The dashed black-red line indicates the detected AR.

measurements became unreliable due to the wet radome of the instrument, and no information on IWV is available for the

following hours. This strong increase in IWV was due to an AR arriving at Ny-Ålesund on 12 January 2018 around 16 UTC. This enhanced water vapor transport from the North Atlantic was stirred by a high-pressure system over Scandinavia and a cyclone developing off the coast of northeastern Greenland (Fig. 12). With the cyclone moving northeastwards, water vapor was transported along its eastern flank to Ny-Ålesund. Continuous, intense precipitation signals of more than 30 dBZ, which only occur less than 1% of the time at Ny-Ålesund (Fig. 8), were observed by the MRR from 22 UTC on 12 January 2018

to 21 UTC on 13 January 2018. With the arrival of the North Atlantic air mass, the 2 m temperature also increased to above 2 °C, resulting in liquid precipitation as indicated by the Parsivel data. For some times during this event, the Parsivel data also indicates solid precipitation. However, these solid precipitation cases are correlated to situations with higher wind speeds and are thus likely misclassified liquid precipitation situations as discussed earlier. With the cyclone slowly moving further northeastwards and polar air being advected at its backside, temperature and IWV decreased again on 14 January 2018, and

precipitation stopped.





Even though a detailed analysis of all extreme precipitation cases will be presented in a future study, we would already like to highlight common features of these precipitation events. Visual inspection of ERA5 output revealed that all these events are related to enhanced water vapor transport from the North Atlantic or Eurasia, often in the form of ARs. The prevailing general circulation patterns feature a high MSLP over Scandinavia/the Barents Sea and/or a low-pressure system located over

the North Atlantic near Iceland. In the case of a (blocking) high-pressure system over Scandinavia, enhanced water vapor transport into the Arctic is realized along its western flank. In the majority of the extreme precipitation cases, cyclones also developed in the Fram Strait or off the coast of northeastern Greenland, which also drive the water vapor transport from the North Atlantic to Ny-Ålesund: water vapor is then advected along their eastern flank, resulting in enhanced precipitation at the site. Additional precipitation might also occur when polar air that is advected on the backside of these cyclones hits the warm

and humid North Atlantic air. These findings are generally consistent with the composite analysis of extreme precipitation events at Spitsbergen by Serreze et al. (2015), who analyzed station and MERRA reanalysis data from 1979 to 2014. They showed that the general synoptic situation is linked to low MSLP systems off the southeast coast of Greenland and between Greenland and Spitsbergen, with positive anomalies in 500 hPa height over Scandinavia and the Barents Sea and negative anomalies centered over Greenland. These conditions favored a southerly flow with advection of water vapor from the North

Atlantic. The strong uplift in the regions of low MSLP also favored the formation of precipitation.

## 5    Conclusions and outlook

Surface observations of precipitation are very scarce in the Arctic. This makes the few locations where continuous precipitation measurements are available even more important. In mid-2017, a MRR, a Parsivel, and a Pluvio were added to the instrument suite at AWIPEV, Ny-Ålesund. Their measurements thus contribute to the existing precipitation observations at Ny-Ålesund,

e.g., the long-term precipitation records by MET Norway. The information of MRR, Parsivel, and Pluvio complement each other so that an overall picture of precipitation amount, type, and vertical structure is achieved. This study has addressed the potential of these observations to characterize precipitation in terms of long-term statistics and individual precipitation events using more than four years of data.

The monthly precipitation totals exhibit a large variability ranging from 1 mm to 155 mm. Considering the effect of un-

dercatch is crucial since it can add more than 100% to the originally measured precipitation value. While a first comparison of Pluvio precipitation amount to the manual daily precipitation measurements by MET Norway revealed a reasonable agreement, an extended comparison including also the Geonor precipitation gauge is planned for the future. Such an analysis could shed further light on the uncertainties in the determination of quantitative precipitation estimates from precipitation gauges in the Arctic. In this respect, the correction functions applied to the precipitation gauge measurements are crucial as pointed out

in Champagne et al. (2024). Following Champagne et al. (2024), a best estimate for the Pluvio data record will be derived. Another aspect is the local variability of precipitation at this complex site. Here, the comparison with the measurements from the Bayelva site (Boike et al., 2018) about 3 km southwest of Ny-Ålesund would be interesting.



Daily precipitation amounts at Ny-Ålesund are typically very small with 50% being smaller than 1.3 mm. Large-scale weather events like ARs and cyclones are common features at Ny-Ålesund and strongly impact the precipitation amount.

While ARs (separated or co-located with other weather systems) occur only 8% of the time at Ny-Ålesund), 43% of the total precipitation amount is measured during these events and 22% during AR events only. Cyclones also play a crucial role, contributing 40% (21%) of the total precipitation amount if all cyclone events (separated cyclone events) are considered.

Determining precipitation type and, thus, the attribution of precipitation amount still poses challenges as well. As no reference data exists, we related the retrieved precipitation type from Parsivel to temperature. Some inconsistencies could be

identified with an increased liquid precipitation frequency at -2°C and an increased solid precipitation frequency at around 5°C. While the latter is related to cases with higher wind speeds affecting the assumption of the fall speed of the particles, the liquid occurrence at low temperatures could not be explained. With the measurements of the video in-situ snowfall sensor, which has been installed in September 2021, these cases can be analyzed in more detail in the future. This might allow for a more refined evaluation of precipitation type from Parsivel or even help to establish an improved (and open source) retrieval

method for precipitation type, which could also directly incorporate temperature information as a further constraint. In many studies, precipitation type is often based solely on a threshold in temperature. However, as indicated by the Parsivel measurements, both solid and liquid precipitation occurs in the temperature regime of -2°C and 4°C. For now, we combined the precipitation type provided by the Parsivel software and the measured 2 m temperature information to allocate precipitation amount to precipitation type. We found that about 22% to 30% of the annual precipitation amount is purely liquid with another

4 to 27% being mixed-phase. The high amount of liquid/mixed-phase precipitation (in total 57%) in 2018 seems unusual. However, a more extended time series is needed to assess the year-to-year variability in the future. Since the performance of the Parsivel degraded in mid-2021, a new instrument was installed in June 2022.

The frequency of precipitation occurrence strongly depends on how precipitation occurrence is defined and on which measurements it is based. While the MRR is very sensitive to a few precipitation particles, resulting in a precipitation occurrence

of 21% (-10 dBZ threshold, 1 min resolved values), Parsivel and Pluvio detect only precipitation during 8% and 1% of the time, respectively. If daily precipitation totals from Pluvio with precipitation amounts larger than 0 mm (1 mm) are considered, precipitation occurrence is 38% (22%).

When looking at individual precipitation events, i.e., times with a MRR signal and precipitation detected by the Pluvio, radar reflectivity values are at least 10 dBZ at some time during the event. This radar reflectivity threshold corresponds roughly to a

rain (snowfall) rate of 4.6 mm h$^{-1}$ (0.6 mm h$^{-1}$). While the quantitative precipitation estimate from the MRR reflectivities was outside the scope of this study, more detailed analyses of radar-based precipitation retrievals will be performed in the future. For example, different $Z_e - R$ and $Z_e - S$ relations, including the proposed method by Schoger et al. (2021) will be applied, and the performance of radar-based precipitation retrievals for Ny-Ålesund will be assessed. The VISSS measurements could also help to constrain the $Z_e - S$ relations further. Such a data set will be valuable for evaluating satellite-based precipitation estimates,

e.g., from CloudSat (Stephens et al., 2002) and the EarthCARE (Wehr et al., 2023) mission, as the MRR also includes vertical precipitation information.



As already seen from the daily precipitation amounts of Pluvio, also the precipitation amounts of the individual precipitation events as defined by MRR and Pluvio are often small, with 50% of these events having a precipitation amount of less than 0.8 mm. Extreme precipitation events, i.e., here the upper 2% of the precipitation events with the highest precipitation amount during August 2017 and December 2021 contribute 28% to the total precipitation amount of all events considered. All the extreme events analyzed coincide with enhanced water vapor transport into the Arctic, with 11 out of 12 cases related to atmospheric river events. The temporally highly resolved precipitation observations by Pluvio, MRR, and Parsivel can capture the temporal development of these events in more detail. Combining the precipitation measurements with the additional information from the other in-situ and remote sensing observations at AWIPEV will thus further shed light on the precipitation processes, e.g., precipitation formation, sublimation, and evaporation. Here, the combination with the cloud radar will be exploited further in the future so that precipitation characteristics can be described in more detail and also be linked to cloud microphysics (e.g., with dual-frequency and polarimetry approaches; Chellini et al., 2023). In addition to detailed case studies, the multi-year data set, which is continuously growing, can be analyzed by exploiting machine learning techniques. These could be used to identify and characterize different precipitation regimes, which are also likely linked to the large-scale synoptic forcing.

Thus, the data set will be very beneficial for evaluating reanalyses, numerical weather prediction, and climate models regarding precipitation in the Arctic. While the strong local scale variability of precipitation at Svalbard might complicate the comparison of the point measurements with coarser resolved precipitation data sets, the high-resolution simulations with ICON-LEM that have been established for Ny-Ålesund (Schemann and Ebell, 2020; Kiszler et al., 2023) allow for a more direct evaluation and will support the interpretation of the measurements in the future.

*Data availability.* The Pluvio (Ebell et al., 2023b), Parsivel (Ebell et al., 2023a) and MRR data (Ebell et al., 2023c) have been published on PANGAEA. Daily precipitation sums of the precipitation gauge of the Norwegian Meteorological Institute (MET Norway) can be found on https://seklima.met.no. 2 m temperature and wind observations at AWIPEV are from Maturilli (2020). The IWV measurements are taken from Nomokonova et al. (2019). The ERA5 reanalysis datasets were provided by ECMWF (Hersbach et al., 2023b, a). The global atmospheric rivers catalog for ERA5 reanalysis is available on PANGAEA (Lauer et al., 2023). The detected weather systems (atmospheric rivers, cyclones, fronts) at Ny-Ålesund for 2017–2021 are available in Lauer (2024).

## Appendix A: MRR calibration evaluation

The absolute calibration of the MRR was evaluated against Parsivel observations of drop size distributions (DSDs) in rain collected during July-September 2022. Radar reflectivity was forward simulated based on the observed DSDs, and compared against values observed by the MRR, following the approach described in Chellini et al. (2022). The calibration offset was estimated to be +0.6 dB, with the positive sign indicating that the instrument underestimates reflectivity. The calibration evaluation was performed based on data recorded outside of the period analyzed in the current study since we noticed biases in the Parsivel data. We observed a systematic underestimation of total accumulated precipitation by Parsivel when compared



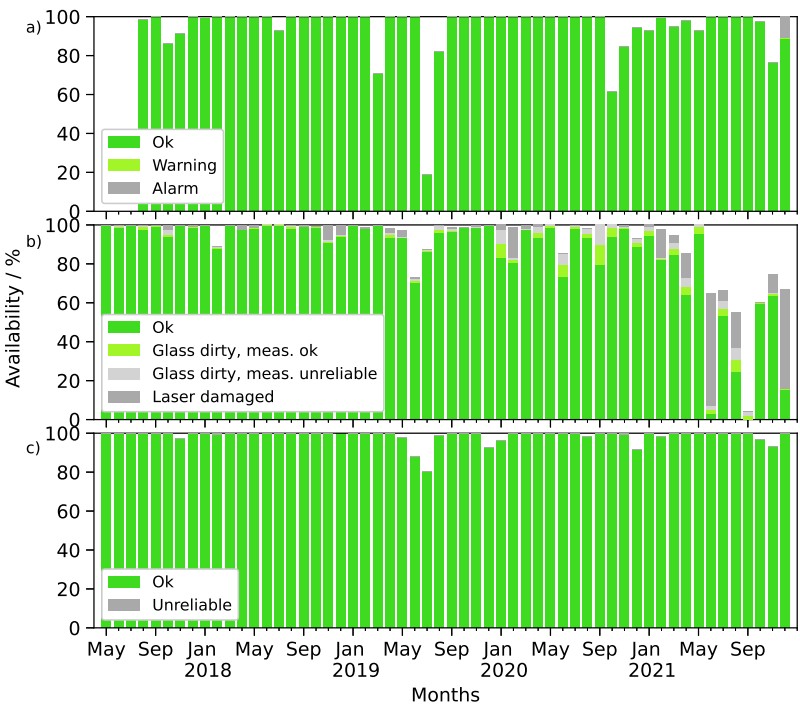

**Figure A1.** Data availability and status of a) Pluvio, b) Parsivel, and c) MRR data (based on measurements at 120 m) from May 2017 to December 2020. Green (grey) colors indicate data that should (not) be used. See legend for more details.

to Pluvio, which might, in turn, lead to biases in the forward-simulated reflectivity used in the calibration. At the same time,
540  we believe that such bias does not affect the precipitation classification used in the analysis. Parsivel was replaced with a new identical instrument in June 2022, which did not display the bias. It is for this reason that we estimate the MRR offset based on data recorded in July–September 2022.

*Author contributions.* KE, RG and MM conceptualized the manuscript. KE and CB analyzed data and prepared the plots. GC and AW worked
545  on processing the MRR data. GC performed the MRR calibration evaluation. PK took care of the instrument operation, data collection and basic processing. ML provided the atmospheric river, cyclone and front detection and helped interpret the results. SD provided visualizations of the reanalysis data and analyzed the figures. KE is the main author of this paper. All co-authors contributed to discussions and reviewed the manuscript.

*Competing interests.* The authors declare that they have no conflict of interest.



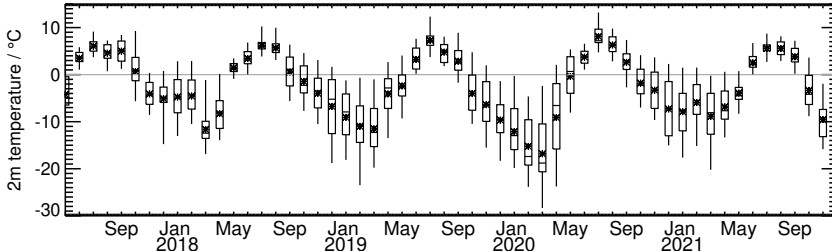

**Figure A2.** Monthly box plots of daily mean 2 m temperature at Ny-Ålesund. The extent of the whiskers indicates the minimum/maximum value. A star indicates the mean value.

*Acknowledgements.* We gratefully acknowledge the funding by the Deutsche Forschungsgemeinschaft DFG (German Research Foundation)
550  - project number 268020496 - TRR 172, within the Transregional Collaborative Research Center "ArctiC Amplification: Climate Relevant
Atmospheric and SurfaCe Processes, and Feedback Mechanisms (AC)3." We thank the AWIPEV team for their support in the operation of
our instruments at AWIPEV. We achknowledge Maximilian Maahn for providing the IMProToo code. We thank Bernhard Pospichal and
Tatiana Nomokonova for installing the Pluvio and Sabrina Schnitt for her support in the data processing.



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
