# Peer review of "Impact of weather systems on observed precipitation at Ny-Ålesund (Svalbard)"

_EGUsphere, 2024_

## Author Comment (AC1)

We thank the anonymous reviewers and Hans-Werner Jacobi for their time in reviewing this manuscript. We very much appreciate the thorough reading and providing us with very constructive comments that improved the present work.
In the following, the answers to the reviewers' comments and questions are given in red. Line numbers refer to the original version of the script and may have changed in the revised version.

**Comments by reviewer 1**

The manuscript entitled "Multi-year precipitation characteristics based on in-situ and remote sensing observations at Ny-Ålesund, Svalbard", present precipitation detection and measurements by several methods in the period 2017-2021 in Ny Alesund and show how atmospheric circulation (AR, cyclones, front) impacts the precipitations events. I think the paper addresses very important questions in an area of rapid change in the context of global warming. The paper is generally well written and the figures are clear and of good quality. However, I think the quality of the paper can be greatly improved by focusing on more specific questions, reorganizing a bit the different parts and expending on the discussions.

First, the introduction is very informative with a lot of cited literature. You introduce well the need of improving ground based observations. However, the connection between the previous studies, the need of new observations for precipitation in Ny Alesund, and the new measurements that you describe is not clear. What are exactly the measurements that were missing before the instruments were installed and how these new measurements can improve our knowledge on Ny Alesund precipitation? the questions that you mention in the introduction: "How much precipitation falls at Ny-Ålesund, and how is it related to the previously mentioned weather systems? What type of precipitation occurs? How often does it precipitate?" have been already addressed in previous studies. You need to point out what these new measurements are adding to the previous answers to these questions and what is new in your way of relating the precipitation events to weather systems. In summary, you need to be more specific on how these new measurements can help answer your research questions.

We have changed the focus of the study and reorganized the manuscript correspondingly. We concentrate now on the Parsivel and Pluvio measurements, leaving out the Micro Rain Radar data since they did not significantly contribute to the study. We expanded the analysis on specific weather events, i.e. atmospheric rivers, cyclones and fronts, and their contribution to precipitation at Ny-Ålesund, which has not been addressed in previous studies yet.
The research questions are now:
- Can the Parsivel constrain a temperature-based mass separation of precipitation into solid and liquid precipitation? How do occurrence and mass fraction depend on temperature?
- How are precipitation amount and type related to large-scale synoptic systems like ARs, cyclones and fronts?
- Which role do these systems play in extreme precipitation events?
We hope that the storyline and the motivation are now clearer.

Since the focus of the paper changed quite a bit we also changed the title to:
"Impact of weather systems on observed precipitation at Ny-Ålesund (Svalbard)"

Second, I suggest reorganizing the manuscript, because in the present form, the flow of ideas is a bit confusing. I really like the part on explaining the weather patterns origins of precipitation events, but it comes too early in the manuscript. I suggest first introducing the results of the measurements (quantity of precipitation, types, frequency), and then the large-scale weather patterns. The advantage of this organization would be that you can use your previous results on separating snow and rain to study the role of weather patterns on rain and snow events. Summer rain, winter rain (rain and snow) and snow, have very different implications for other research studies and local communities, so it would be extremely interesting to use the pluvio and parsivel measurements to study the specific origins of rain and snow. This analysis would be an application of your method of separating rain and snow. Regarding the part on frequency and the use of MRR, I don't really see the benefit on this study. It needs to be better explained or to be removed from the analysis.

We have reorganized the manuscript. We present first the measurements and their uncertainties in more detail before focusing on the weather system analysis. As suggested, we also removed the MRR analysis as it did not add much further information in the way we incorporated the data. We will exploit the MRR measurements in more detail in future studies.

Finally, your article lacks a discussion part where you talk about previous studies and show how the new measurements may improve the previous studies. Many articles are already cited in the introduction but they don't serve your discussion in the current version of your manuscript. You have some discussion spread in the manuscript, but it would be good to write a specific discussion section and expend on them.

By changing the focus and reorganization of the manuscript, we hope that the link to previous studies is also clearer now. We tried to better incorporate the discussion parts. We find that it is more natural to discuss the results when they are presented. This is why we do not have a separate discussion section. However, we also extended the discussion in the summary and conclusions section since this also directly motivates the needs and ideas for upcoming studies.

I think this manuscript have a great potential to get published after major revision, and is a very important step in improving our precipitation knowledge in Ny Alesund.

Thank you

Here are some specific comments:

I suggest changing the title to "Multi-year precipitation characteristics based on in-situ and ground-based remote sensing observations at Ny-Ålesund, Svalbard".

Since we changed the focus of the manuscript we changed the title to:
"Impact of weather systems on observed precipitation at Ny-Ålesund (Svalbard)"

l.11 «Cyclones contributed 40% (21%) of the total precipitation amount if all (separated) cyclone events are considered ». This sentence is not clear to me.

We rewrote the abstract. The sentence now reads, "ARs occurred only 8% of the time at Ny-Ålesund but contributed to about 42% of the total precipitation amount with a high liquid mass fraction (72%)."
Note that the numbers slightly changed since we found a coding error in reading the Pluvio data.

l.13-14: I am not sure this is needed. Of course the occurrence is lower when the resolution is 1 minute compared to daily. What is the resolution of micro rain radar?

We removed this result from the abstract but included it in the instrument section. We think that it is important to understand the sensitivities of the different instruments which might also affect statistics as precipitation occurrence.

l. 32-40 maybe separate what is observed in the past and what is modeled in the future, it is bit mixed up here. There is already literature on past increase of precipitation and rain in the Arctic that you can include.

We revised this section and first mention studies of detected precipitation changes using observations and reanalyses. At the end of the section, we refer to the results of climate projections.

l. 60-61 «melt days in winter in Svalbard and the associated precipitation sums have increased» not sure I understand what are precipitation sums and how it is associated to melt.

We rephrased the sentence:
"Vikhamar-Schuler et al. (2016) further showed that the occurrence of melt days, i.e. days with temperature >0°C, and the accumulated precipitation during these events have increased in Svalbard in winter."

l. 61 «These rain-on-snow events». I would say «the rain on snow events» since you don't talk about rain-on-snow events previously.

We rephrased the sentence:
"Rain-on-snow events, which have implications for the cryosphere, ecosystem and infrastructure, have also been studied in further detail (e.g., Hansen et al., 2014, 2019; Peeters et al., 2019; Xie et al., 2024)."

l.77-78: this sentence can be simplified. you say three time the same thing: «model data» «from numerical weather prediction models»and «climate simulations».

We removed this paragraph in the revised version.

l.104: operated by?

corrected

l.140 What is the resolution of parsivel?

We added this information:
"The OTT present weather sensor Parsivel2 is an optical laser disdrometer. It provides information on fall speed, size and type of precipitating particles in 1 min temporal resolution."

l.164: what is the horizontal resolution of the MRR?

We removed the MRR from the manuscript. To answer your question:  The antenna opening angle (beam width at half maximum) is 1.5°. This means that the horizontal resolution is about 3 m at 120 m height, and 26 m at a height of 1 km.

l.188-189: you can just say june to september

 Changed to:
"In general, daily mean T2m values are above 0°C from June to September and rarely exceed 10°C (Fig. A2)."

l.231 it seems that the underestimation of your pluvio data compared to MET norway is larger in winter months. It would be interesting to also show the average difference per month.

We included also a time series with the monthly differences in Fig. 4 and scatter plots of monthly and daily values in Fig. A3, where we also provided bias, RMSD, standard deveiation and correlation. We discussed the differences in more detail and also incorporated the corrected MET Norway precipitation values (ensemble correction) by Champagne et al. (2024) as suggested. See new section 3.1.

l. 232-23 you can use the precipitation correction of the MET Norway time series by several methods here https://www.easydata.earth/#/public/metadata/a3d7b9e6-9626-4d43-bb83-623900eb1053. It would be good to include them in table 1 and figure 2. I suggest also to show a scatter plot with corrected MET norway vs corrected pluvio for daily values.

 See previous reply.

l.257: methodology of?

"by" is also possible

Table 2: AR-CY not AR-CA. The results are strongly dependent on the occurrence of these weather patterns. It would be good if overall occurrences appear somewhere.

We have added a column with the total occurrence from August 2017 to December 2021 in Fig. 7 and explicitly mention these values as well as the yearly values in Table 4.

l.259: you mean a monthly maxima?

Yes, we corrected it to "monthly maxima".

Figure 4: it also seems that the combined AR-CY don't happen frequently but are associated to a lot of precipitation. I think this needs to be discussed.

We highlight this point in the revised version:
"Even if the occurrence of ARs is rather low on average (4% for O-AR, 8% for all ARs), they contribute 22% (O-AR) and 42% (all AR) to the total precipitation, respectively. The relatively rare combined classes AR-FR (2%), AR-CY (1%) and AR-CY-FR (1%), contribute together 20% of the total precipitation amount."

l.264-266: I think you should discuss the specific months after discussing the general behavior.

We changed the order to discuss first the general behavior.
"The largest contributions to monthly and yearly precipitation can be found for the AR and cyclone classes. Even if the occurrence of ARs is rather low on average (4% for O-AR, 8% for all ARs), they contribute 22% (O-AR) and 42% (all AR) to the total precipitation, respectively. The relatively rare combined classes AR-FR (2%), AR-CY (1%) and AR-CY-FR (1%), contribute together 20% of the total precipitation amount. However, the year-to-year and month-to-month variability of the precipitation fraction associated with ARs is large, with only 25% in 2021 and even 50% in 2018. In particular, in the very wet month of September 2017, almost all precipitation, i.e., 145 mm (Fig. A5), can be related to the (co-)occurrence of ARs. For the month with the highest precipitation amount, i.e., November 2020, both AR and cyclone classes contribute together to about 80% of the total precipitation amount, with the O-CY class even dominating."

Figure 5: it would be better to write the bins in th x axis: [-0.5-0.5], [0.5-1.5], [0.5-1.5]... because I am not sure what bin corresponds to each point here. For example, does 0 corresponds to [-0.5-0.5]?

Since the space along the x-axis is not sufficient, we added this information in the caption of Fig. 5 and explicitly listed the values of the "phase transition regime" in Table 2.

l.276: I question the choice of including hail in solid precipitation as snow and hail are driven by very different processes and may not occur in the same season.

We have added a comment on this in the text:
"We included "graupel" and "hail" in the solid class even though the microphysical processes might be quite different in these cases. However, the occurrence of these two classes is very low (<1.9% for graupel and <0.001% for hail) and does not impact the key findings."

l. 280 I think MET Norway also gives the type of precipitation: snow, liquid, mixed. It may be good to compare with this. And also to compare with snow-rain separation done by Champagne et al. 2024 using a threshold of 1°c (dataset above). I think your study as the potential to determine more precisely a threshold for solid vs liquid precipitations that can validate previous datasets or be used in future studies. It is great and can be emphasize in your manuscript.

We included a discussion on using a threshold of 1°C as used in Champagne et al. (2024) in the analysis. See also Fig. 6b and Table 3.

lines 320-331
"We also analyzed the effect of using a simple temperature threshold (TS1°C) assuming all precipitation to be solid for temperatures <1°C as in Champagne et al. (2024) (Fig. 6b). For some months, this significantly increases liquid precipitation (by up to 53 mm) resulting in generally higher yearly liquid precipitation fractions with anadditional six to 15 percentage points (Table 3). Using only the temperature-based mass separation (TMS) as derived from the Parsivel observations (and thus no direct Parsivel observations at all), has a smaller effect, even though for some months differences are several millimeters showing still the uncertainty related to phase attribution. However, the yearly liquid mass fraction of the TMS method is similar to the combined Parsivel/TMS method (Table 3).
Since Champagne et al. (2024) applied the 1°C temperature threshold to hourly mean 2 m temperature values, we also calculated hourly liquid and solid precipitation sums from the 1 min resolved liquid and solid values of the combined Parsivel/TMS method and set those in context to hourly mean 2 m temperatures (Fig. A4). Also, for hourly averaged 2 m temperatures and hourly accumulated liquid and solid precipitation sums, we find a similar temperature dependency for the mass separation as shown in Fig. 5b."

However, we did not use any further MET Norway data as we just used the data set provided by Jacobi et al. (2024). However, it is definitely interesting to look further into the phase separation of other data sets in the future. As for the Geonor data, we would postpone this to a later stage since this should be done in collaboration with MET Norway.

l.283 «this makes sense» is a bit familiar. Temperature is mostly used because there is no alternative and because temperature and precipitations are often long records measured together. In my opinion, what is very relevant your study is that the temperature threshold can be validated.

This paragraph is obsolete in the revised version.

l.285: When you say 2°c you mean the 1.5c-2.5°c bin? Since you use a 1minute resolution, with a lot of data available, I would suggest using more bins, at least for critical temperature between 0 and 3°c (0.1°c wide bins?). You can then arrive with a suggestion of thresholds (or a ranged threshold) separating solid, liquid and mixed precipitation.

We revised the phase discrimination in the revised version, distinguishing only liquid and solid precipitation. See section 3.2. The temperature bins have been refined to 0.2°C.

l.293-294: Not sure I understand when you say «during periods when solid precipitation has been detected by Parsivel as well (not shown)». parsivel have shown mixed precipitation during these events then?

Parsivel has detected in some minutes solid precipitation and in some other minutes liquid precipitation.
We rewrote the sentence:
"Looking at these cases in more detail reveals that all these situations occur during periods when solid precipitation only has been detected by Parsivel in other minutes (not shown)."

l.294: what the size of particle infers for the type of precipitation? Expend on that.

We rearranged this paragraph so that the reasoning is clearer now:
"Visual inspection of the pictures of the particles taken by VISSS for a case on 5 May 2023 showed that only solid precipitation was present (Maximilian Maahn, University of Leipzig, personal communication 25 August 2023). Interestingly, Chellini et al. (2022, 2023) found that low-level mixed-phase clouds at Ny-Ålesund produce small fast-falling ice particles in this temperature regime, which could be misinterpreted as drizzle. The detected Parsivel particle sizes are relatively small during these cold "liquid" events, with a mean volume equivalent diameter of 1.3 mm only. We thus assume that the Parsivel algorithm falsely classifies these smaller solid particles in this temperature regime as "rain" or "drizzle"."

l.314-315: I think the thresholds for mixed precipitation can be refined. Looking at figure 5, more than 90 % of precipitation is only solid below 0°c and only liquid above 3°c. Keep in mind that for most studies mixed precipitation is a useless variable. For most fields, we must know if snow is falling and sticking to the ground or if the precipitation are liquid (or almost liquid at the surface). The Parsivel is a very powerful tool that should be used at its best capacity to refine the temperature threshold between liquid and solid precipitation.

We agree that a mixed-phase precipitation class is not helpful here. We revised the classification procedure. See also Section 3.2 for more details.

"We took all corrected 1 min resolved Pluvio precipitation values larger than 0 mm into account, for which also the Parsivel had detected a precipitation signal within ±10 min. The Pluvio precipitation signal was then declared as solid if the classes "snow", "snow grains", "graupel" and "hail" were the dominating precipitation types within the ±10 min interval. We included "graupel" and "hail" in the solid class even though the microphysical processes might be quite different in these cases. However, the occurrence of these two classes is very low (<1.9% for graupel and <0.001% for hail) and does not impact the key findings.
If the liquid Parsivel classes "drizzle", "drizzle with rain", or "rain" was dominating, the Pluvio precipitation amount was associated with liquid precipitation. In a few cases (0.7% of all cases), mixed-phase precipitation ("rain, drizzle with snow") was dominating the Parsivel signal. Here, half of the Pluvio precipitation amount was attributed to solid and half to liquid precipitation. However, since these cases contribute only 0.7% to the total precipitation amount they do not significantly affect the results. The occurrence of liquid and solid precipitation was then analyzed as a function of 2 m temperature.
[...]

To split precipitation into solid and liquid for the whole period August 2017 to December 2021, we applied a combined Parsivel/temperature-based mass separation (TMS) method: for temperatures <0.2°C, we assume all precipitation to be solid. All precipitation is assumed to be liquid for temperatures ≥3.6°C. For the temperature range in-between, we check first if Parsivel detected precipitation within ±10 min and if wind speeds are <5 ms$^{-1}$. If this is the case, we use the Parsivel classification, as explained earlier, to discriminate between liquid and solid and attribute the precipitation mass correspondingly. If precipitation phase information is not available from Parsivel due to no or no valid Parsivel data (in particular in 2021; cf. Fig. A1b), due to no detected precipitation by Parsivel, or due to wind speeds ≥5 ms$^{-1}$, the 2 m temperature is used for the mass separation as shown in Fig. 5b) (for the exact values see Table 2). In some cases, no temperature measurements were available, so the precipitation phase could not be determined for the corrected Pluvio precipitation amounts. However, this affected less than 2 mm of the whole precipitation amount in the period from August 2017 to December 2021."

Figure 6: why for mixed phase the fraction of precipitation occurrence is way lower than the fraction of precipitation amount?

This part has been removed.

You could also look at the dataset from Champagne et al. 2024 https://www.easydata.earth/#/public/metadata/a3d7b9e6-9626-4d43-bb83-623900eb1053 and validate if a simple way of separating snow and rain (temperature threshold) can be done. your study can be very useful to validate such dataset.

We also checked the sensitivity of the liquid/solid partitioning by applying also the 1°C temperature threshold as used by Champagne et al. (2024). For the final liquid/solid precipitation sums in Champagne et al. (2024), differences might additionally arise due to the partitioning of solid and liquid precipitation of the 12 hourly resolved data using 12 hourly mean temperatures. We agree that it is interesting to compare this in more detail but would also postpone this analysis to a follow-up study.

l.323: «Since the mixed-phase precipitation type, as detected by Parsivel, only seldom occurs» awkward wording.

This part is obsolete.

l.326: I think you can reduce the number of unknown cases by narrowing down the threshold [-2-4°c].

Combining the Parsivel measurements with temperature information reduces the unknown cases.
"In some cases, no temperature measurements were available, so the precipitation phase could not be determined for the corrected Pluvio precipitation amounts. However, this affected less than 2 mm of the whole precipitation amount in the period from August 2017 to December 2021."

l. 333 i don't see any analysis on ROS events.

We added an analysis of "liquid precipitation days". We actually did not analyse rain on snow events since we are not sure about the surface state during the liquid precipitation event. See section 4 and Fig. 11

"Another type of "extreme" precipitation event is liquid precipitation during the cold season. As mentioned before, rain-on-snow events are of particular interest since they can have severe implications for wildlife and Arctic communities. We investigated the number of days with liquid precipitation >1 mm in each month and connected it to the occurrence of the different weather systems (Fig. 11). As expected, most of these days can be found from May to September when temperatures are predominantly above 0°C (Fig. A2). However, except for the relatively cold 2019/2020 winter (Fig. A2), liquid precipitation days are also common from November to April. Almost all liquid precipitation days are connected to at least one of the weather systems and all liquid precipitation days from November to April (22 in total). 91% of these days are connected to ARs and 64% and 45% to fronts and cyclones, respectively."

l.345: >0mm or 1mm ?

This part has been remove in the revised version.

l.348-349 «To understand which rain or snowfall rate a value of -5 or -10 dBZ actually represents, Ze − R and Ze − S relationships must be applied » Not clear. Reformulate.

This part has been remove in the revised version.

l.370: Even though these frequencies serve different purposes, I think you still need to discuss a bit if these frequencies are plausible or not. The MRR frequency seems very high, but pluvio rather low. what causes the overestimation in MRR and Parsival? Also expend on the purpose of these different measurements (maybe in a separate discussion part).

The MRR has been removed from the analysis. We moved this part to the instrument description as it is important to understand the different instrument sensitivities.

"Fig. 2b shows the frequency of detected precipitation by Pluvio. Using the 1 min resolved Pluvio time series results in monthly precipitation frequencies of up to 5% only and in all-time average values of 1%. Using daily accumulated Pluvio data increases the monthly precipitation frequency to 4%–63% and, if a threshold of 1 mm is applied, to 4%–46%. […]
Compared to the 1 min resolved Pluvio measurements, the precipitation signal occurrence is much higher for Parsivel (Fig. 2). For the whole period (August 2017 – December 2021), it is 8% (compared to the 1% of Pluvio). This is due to the fact that the Parsivel already detects a few precipitating particles whose mass might not be large enough to be measured by the Pluvio."

Table 4: I assume these are data from pluvio? Are these raw or corrected data? It would be nice also to see the data from corrected Norway MET data."

We added the corrected MET Norway data in Table 6 (table number changed). We simplified the extreme event analysis and analyzed daily values (Table 6 in the revised version). In the future, we will have a closer look at the temporal development of these events, exploiting also the various remote sensing observations. We clarified in the table caption that we use the corrected Pluvio data. We did not add the MET Norway data here since we analyze daily precipitation sums from 0-24 UTC. Comparing it with the 12 hourly resolved MET Norway data (06-18 UTC, 18-06 UTC) would result in sampling differences.

l.375-386: What is the main benefit of this MRR for precipitation detection? I don't really see it here. You are not really describing the vertical variability of the results and how it can be used. This needs to be discuss more.

We removed the MRR from the revised version.

l.393 not clear. You mean you need to have at least 30 minutes with a Ze < -10dBZ (no precipitation) between two events?

We remove the MRR from the revised version.

l.394: an event interrupted by for example 12h without precipitation is also likely from the same weather system. So the 30 minutes needs to be better justified.

We simply used daily precipitation sums for the event analysis and changed the manuscript accordingly.

l.395 discussed later in another paper you mean? I don't see this discussion.

This part is obsolete in the revised version.

l.396-397 «Here, we extended the considered period by ±10 min due to the spatial distance of the Pluvio and MRR measurements» you mean that rain may take 10 minutes to reach one site to another? This needs to be better explained.

This part is obsolete in the revised version.

l.403-406 I don't really understand the goal of figure 9.

This part is obsolete in the revised version.

l 410: «maximally contribute to the total precipitation amount with 11%» not clear

This part is obsolete in the revised version.

l.407-421: The results are interesting but I question what is the end goal of showing all these results. The longer and stronger events of course are more rare but give more precipitation amounts. What is new about it? What are the implications of these results? More generally I have hard time to see the benefit of using MRR here.

This part is obsolete in the revised version

l.424 A follow up study would be very interesting as your article leads to lot of new questions. I think it would be very interesting to separate the type of extreme events (summer rain, snow, rain on snow) and to focus on ROS events. It can be done here and would show a good application of pluvio-parsivel.

We added some statistics on liquid precipitation days, i.e. days with liquid precipitation >1 mm and connected them to the weather systems. See reply to comment l. 333.

l.464-472 I would first summarize your main results before introducing what may be done in the future

We rewrote the summary and conclusion section:
First, the results of the precipitation phase partitioning method are summarized followed by a discussion and outlook in this respect.
Then, the results of the weather system analysis are summarized, discussed and also here an outlook given.
We hope that the structure is clearer now.

l.473 replace small by low

Replaced.

l.483-485 I think from your results you can already give approximate temperature threshold for rain, snow and mixed precipitation at Ny Alesund. This is extremely useful for future studies and this needs to be higlighted in your article.

We highlighted this aspect in the paper and also formulated a corresponding research question:

- Can the Parsivel constrain a temperature-based mass separation of precipitation into solid and liquid precipitation? How do occurrence and mass fraction depend on temperature?

I suggest adding a discussion section and the conclusion needs also to be rewritten with more emphasis on very important results. In the current form the conclusion repeats a lot of previous results but it is hard to see what are the main conclusions.

We rewrote the conclusions part and hope that the main results are now presented more concisely.

**Comments by reviewer 2**

**General Comments**

The study investigates precipitation characteristics at Ny-Ålesund, Svalbard, based on ground-based observations from three instruments (Pluvio, Parsivel and MRR), and relates

precipitation to weather systems such as atmospheric rivers, cyclones, and fronts. It also investigates extreme precipitation events and describes one precipitation event in more detail.

The manuscript is generally well written and logically structured, and presents a comprehensive dataset. The data analysis is mainly sufficient, figures are of good quality, and conclusions are supported by the data presented. However, there are some areas which need major improvements, particularly the clarity of the study's motivation, the description of methods, and discussion of implications, see my specific comments. The text should be substantially clarified and improved. With improvements, this study will make a valuable contribution to understanding precipitation in the Arctic.

We have changed the focus of the study and reorganized the manuscript correspondingly. We concentrate now on the Parsivel and Pluvio measurements, leaving out the Micro Rain Radar data since they did not significantly contribute to the study in the way we incorporated the data. We expanded the analysis on the weather events, i.e. atmospheric rivers, cyclones and fronts, and their contribution to precipitation at Ny-Ålesund, which has not been addressed in previous studies yet.
The research questions are now:
- Can the Parsivel constrain a temperature-based mass separation of precipitation into solid and liquid precipitation? How do occurrence and mass fraction depend on temperature?
- How are precipitation amount and type related to large-scale synoptic systems like ARs, cyclones and fronts?
- Which role do these systems play in extreme precipitation events?

We hope that the storyline and the motivation are now clearer.

**Specific comments**

Introduction:

1. The introduction gives a strong rationale but it is quite general. It should better connect the objectives of this study to gaps in the literature.

We adapted the introduction and formulated clearer objectives of the study resulting in the research questions mentioned above.

Methods:

2. The correction function for precipitation by Wolff et al (2015) is based on measurements made at Haukeliseter, Norway (at above 1000 m altitude). How well does this correction function apply to the conditions in Ny-Ålesund? How did you select this correction function? There are plenty of other correction functions in the literature (see for example Kochendorfer et al. 2017, or Køtzow et al. 2020). Discuss the assumptions or limitations of the selected function and motivate the selection briefly.

Kochendorfer, J. et al: Analysis of single-Alter-shielded and unshielded measurements of mixed and solid precipitation from WMO-SPICE, Hydrol. Earth Syst. Sci., 21, 3525–3542, https://doi.org/10.5194/hess-21-3525-2017, 2017.

Køltzow, M. et al: Verification of Solid Precipitation Forecasts from Numerical Weather Prediction Models in Norway. Wea. Forecasting, 35, 2279–2292, https://doi.org/10.1175/WAF-D-20-0060.1. 2020

We are aware that many correction functions are available. We have chosen the one by Wolff et al. (2015) since it does not require a phase partitioning of the precipitation amount into solid, liquid or mixed before. We mention this explicitly also in the text:

"We also applied an empirical correction function by Wolff et al. (2015) to the 1 min precipitation data to correct for wind-induced precipitation losses. This correction function has been developed based on gauge measurements in southern Norway and depends on temperature and wind speed at gauge height (see Eq. 12 in Wolff et al., 2015). The advantage of this correction function is that it can be directly applied to the total precipitation amount and does not require a mass separation of the precipitation into liquid and solid first. As this paper does not focus on evaluating correction functions, we made a choice here but want to point out that the estimated undercatch strongly depends on the chosen correction function (Champagne et al., 2024)."

We also discuss the impact on the correction function in the new section 3.1.
From the comparison with the uncorrected MET Norway precipitation gauge data, we deduce that the correction function by Wolff et al. (2015) likely still underestimates the precipitation amount. As the present study does not focus on the evaluation of correction functions, we plan to have a closer look at different correction functions in the future. Here, we would also like to incorporate the Geonor measurements as they rely on the same measurement principle as the Pluvio and also experience similar measurement conditions (since the two instruments are only about 140 m apart). We added a corresponding paragraph also in the outlook section:
"Still, a few points should be noted regarding the presented analysis. The absolute values of the precipitation amount are still uncertain. As seen from the comparison with the uncorrected Geonor data, the corrected Pluvio measurements (using the algorithm by Wolff et al., 2015) are likely still underestimated. Following Champagne et al. (2024), different correction functions will be applied in the future to better account for the uncertainties of the Pluvio data record. An extended comparison with the processed Geonor precipitation data will provide further insight into the measurement uncertainties."

3. Clarify which instruments are used for specific characteristics (precipitation amount, frequency, and type). For instance, in Section 2.3, it is currently unclear why MRR data is used or how it complements those from Pluvio and Parsivel.

We reorganized the instrument section. The MRR data are not part of the study anymore.

4. The abbreviations O-AR, O-CY, etc., are introduced in Section 2.4 but not clearly explained. I suggest briefly clarifying these terms when they are first mentioned (O-AR refers to 'only atmospheric rivers'). Additionally, as the use of ERA5 data and the detection of weather events are central to the analysis and conclusions, you should consider creating a dedicated

subsection to explain the detection algorithms and their relevance to the study. This would enhance clarity.

We introduced the abbreviations in section 2.4, which is dedicated to the detection algorithms. We also expanded the explanation of the weather system detection as suggested.

5. Why was Geonor not used in this study? Wouldn't it provide an additional comparison or insights into precipitation measurement uncertainties?

We planned to include the Geonor in the revised version. However, when discussing with Mareile Wolff, we figured out that the processed Geonor data are available for a short period only. The "raw" Geonor data, i.e., the bucket content, still needs to be translated to precipitation amount, for which corrections and quality checks also need to be applied (checks which are already included in the Pluvio software). The precipitation amount is not simply the difference of the bucket content at time step i+1 minus the bucket content at time stamp i. So, there is not a ready-to-use Geonor data set available. More sophisticated processing steps need to be applied to be able to use the data properly. This is definitely the next step that we would like to take but this requires more time.

Section 3

6. Small precipitation amounts are the focus of Figure 3 and the paragraph starting on line 240. The importance of this analysis is not clearly motivated. Please explain why studying these small precipitation events is relevant.

We moved this discussion to the section on the weather types (section 4) and hope that the motivation is now clearer.
"Very small precipitation amounts or trace precipitation, i.e., small but immeasurable daily precipitation events, are still challenging for observations and models. Boisvert et al. (2018), who defined trace precipitation as days with less than 1 mm precipitation, showed large differences in the occurrence and annual amount of trace precipitation over the Arctic Ocean between eight reanalyses. However, trace precipitation can make up a substantial proportion of the total precipitation amount over the central Arctic Ocean (Boisvert et al., 2018). The question of whether these small amounts of precipitation that numerical models frequently generate occur also in reality has not yet been completely answered. This is also due to missing accurate reference observations. At Ny-Ålesund, trace precipitation (i.e., non-zero daily precipitation amount <1 mm) is reported from the corrected Pluvio data for about 16% of the time of the analyzed period. It accounts for 44% of the days with precipitation recorded. Trace precipitation is thus a common feature of the atmospheric state at Ny-Ålesund. The annual trace precipitation amounts for 2018-2021 are between 20 to 30 mm. Compared to the annual precipitation amount, these values are rather small. For example, for 2021, the annual trace precipitation amount is 5.5% of the total precipitation amount at Ny-Ålesund. Days with trace precipitation can be mainly related to the residual class (43%) followed by cyclone-related events, in particular with the O-CY class (18%). Focusing more on processes on the local scale, trace precipitation could also be associated with the frequent occurrence of low-level mixed-phase clouds in conjunction with katabatic winds (Gierens et al., 2020), with the dry katabatic flow leading to the sublimation of a large portion of the precipitating mass."

And in the conclusions section:

"As these models often produce a lot of small, potentially artificial precipitation amounts, it would be interesting to look more into the trace precipitation events. While these events are probably the most challenging ones for precipitation gauge observations, in particular for classical manual gauges, the higher sensitivity of the Parsivel might be beneficial. Also, additional observations from the cloud and micro rain radar will be helpful in identifying blowing snow events that might be falsely interpreted as precipitation."

7. Lines 290–302 discuss mixed-phase precipitation but do not address sub-zero liquid precipitation, which also occurs. Is this considered in your analysis?

As the first reviewer mentioned that a mixed-phase precipitation class is not helpful for many studies, we revised our phase discrimination strategy to come up with solid and liquid precipitation classes only. Please see also the reply to the comment of reviewer 1 with respect to the lines 314-315.

The Parsivel does not indicate a lot of sub-zero liquid precipitation (see Fig. 5). We think that the enhanced liquid occurrence around -3 to -2 °C is an artifact (as discussed in section 3.2). This is why we do not assume liquid precipitation for sub-zero temperatures.

8. Could additional data sources validate precipitation types from Parsivel? For instance, the video data you mention or manual observations (SYNOP/METAR)?

To further incorporate the measurements of the video in-situ snowfall sensor (VISSS) is definitely the next step. It will be interesting to see how well this data can be used to evaluate the Parsivel precipitation classes. However, VISSS measurements started only in the autumn of 2021. We are thus planning to use the data in the evaluation for the more recent years.

SYNOP data might be more challenging to use due to the different temporal resolution and thus, representativity of the observation. We are not sure if such a comparison will provide us with more insights.

Currently, a Thies disdrometer is operated by the University of Leipzig near the balloon hall about 30 m away from the Parsivel. The comparison between these two instruments will help us to better understand the performance of the disdrometers, at least to understand how consistent the measurements of the two instruments are. The VISSS data can also here be used as a reference. We added a paragraph in the outlook section in this respect:

"With the measurements of the video in situ snowfall sensor, which was installed in September 2021, these cases can be analyzed in more detail in the future. This might allow for a more detailed evaluation of precipitation type from Parsivel or even help to establish an improved (and open source) retrieval method for precipitation type, which could also directly incorporate temperature information as a further constraint. In February 2025, a Thies disdrometer was operated by the University of Leipzig close to the balloon hall about 30 m away from the Parsivel. A comparison of the detected precipitation and precipitation phase will shed further light on the accuracy of the disdrometer-derived precipitation phase classification."

9. Line 346: How was the height of 120m selected? Can precipitation detected at this height reliably represent surface precipitation? What uncertainties are associated with this approach? Discuss , and modify the methods section accordingly.

Obsolete since removed in the revised version.

Section 4

10. Line 401: "for which Ze values precipitation is actually detected on the ground" is a critical question that applies to previous sections as well (e.g., precipitation frequency analysis). Please address this earlier in the text.

Obsolete since removed in the revised version.

11. Lines 401–406: Results are described but not explicitly interpreted. Connect these findings to the question you are trying to answer. For example, explicitly address how these Ze values relate to surface precipitation detection.

Obsolete since removed in the revised version.

12. Figure 10: I am not sure about the journal recommendations for figures, but for me, it would be easier if a legend was added directly in the figure, not just the caption. This applies to other figures as well, but it is especially important here because of the number of lines in the plot.

We revised the figures in general and added further legends to the plots.

13. Captions in general: Indicate which data sources the figures are based on. While this becomes clear when reading the text, captions alone should provide this information.

We revised the figure captions and included all necessary information.

14. The analysis of extreme precipitation cases is valuable, but the manuscript refers too much to future studies. For example, on line 441, avoid starting with a reference to a follow-up work. Focus on the results gained from these cases now, such as the role of ARs.

We revised the analysis and wording and focused more on the results presented. Please see also the corresponding section in the revised version (last 2 paragraphs of section 4).

Conclusions:

15. Clearly state the main conclusions. Right now, they are mixed with comments about limitations and future work, which makes it hard to focus on the key messages. I suggest putting limitations and future work into separate paragraphs.

In the conclusion section, the results are now presented in a more concise way:
First, the results of the precipitation phase partitioning method are summarized followed by a discussion and outlook in this respect. Then, the results of the weather system analysis are summarized and discussed and an outlook is given.

16. Line 473: The statement "Large-scale weather events like ARs and cyclones are common features at Ny-Ålesund" appears to contradict the statement on lines 473–474 about ARs occurring only 8% of the time. Please clarify.

This statement has been removed.

17. Line 486: The reference to "In many studies" is vague. Specify which studies base precipitation type solely on temperature thresholds.

We added corresponding references.

18. Discuss how your findings contribute to current knowledge about Arctic precipitation, as introduced earlier in the manuscript. How do they advance our understanding of precipitation processes in the Arctic climate? I would like to see more discussion on what these results mean and what the community can learn from this study.

We revised the introduction and summary and conclusions sections. We hope that the new aspects and the key results of the study are much clearer now. We would refer the reviewer here to the corresponding sections 1 and 5.

19. Specifically, you mention model uncertainties in the introduction. Could your observations help improve understanding and even improving models? What are the potential applications for the modeling community?

The additional Pluvio observations will help to better describe the uncertainties in gauge-based precipitation observations. These uncertainties are crucial when observations are compared to model output and differences interpreted. Comparing the precipitation observations of the various locations in and outside of Ny-Ålesund will also help to better understand the local variability of the precipitation measurements and the representativity of these observations keeping the horizontal resolution of the model in mind.

Two specific aspects might also be of particular interest: the phase partitioning in models (and the dependency on temperature), as well as the frequent occurrence of trace precipitation in models. Currently it is unclear if trace precipitation events in models are artifacts or not. Here, the Pluvio and Parsivel measurements can contribute with more detailed observations having a higher sensitivity as traditional manual precipitation gauges.

We have added a paragraph on the importance of the precipitation phase, the temperature dependency and current limitations in the in the introduction. Trace precipitation is addressed in section 4 in more detail. We come back to these two topics also in the conclusion section.

**Minor issues**

Line 27: "the Svalbard archipelago … reveals the highest temperature increase" sounds strange. → The Svalbard archipelago exhibits, or has experienced.

We corrected the verb to "has experienced".

Figure 2 caption: Clarify that the hatched areas denote months with significant data gaps.

We added this information to the figure caption.

Line 475: Remove the extra ) after Ny-Ålesund.

changed

**Community Comment by Hans-Werner Jacobi:**

Dear authors:

I stumbled over the following statement in the introduction: "A recent study by Zhou et al. (2024) revealed that the Arctic warming between 1979 and 2001 is three times higher than the global warming." I think this is a misinterpretation of the Zhou et al. paper. First of all, they analyse data (observations and model results) for 30 year periods or longer. I could not find any information on the mentioned 22 year period from 1979 to 2001. Moreover, Zhou et al. calculate from observations a 3.98 faster warming for the period from 1980 to 2014, which increased even further for more recent period (see their Figure 1a). In my opinion, there is no doubt that since ~1980 the observed warming in the Arctic was approximately four times faster than the global warming as already demonstrated in other studies. By comparing with model results, Zhou et al. further conclude that the Arctic Amplification should lead to an "only" 3 times faster warming in the Arctic and that the recent accelerated warming in the Arctic is related to natural variability. This means that in the long term the accelerated warming in the Arctic can be expected to fall below a factor of 3. However, this does not undermine that in the last 40 years the observed warming in the Arctic was ~4 times faster than the global warming.I think the initial statement in the introduction rather leads to confusion and should be rectified.

Citation: https://doi.org/10.5194/egusphere-2024-3368-CC1

Dear Hans-Werner, thank you for pointing out this mistake. The information from the two papers has been mixed up incorrectly. We changed the sentence to:

"Recent studies have shown that Arctic warming during the last decades was four times higher than global warming (Zhou et al., 2024; Rantanen et al., 2022)."

---

## Referee Report (RR1)

**General comments**

The authors have substantially improved the manuscript by removing the section on the MRR, reorganizing the text, clarifying the focus, and improving the figures. The responses to the reviewer comments are also excellent. I really appreciate the careful work by the authors, and I am pleased with the changes made to the manuscript. The paper now reads very well, and I truly enjoyed reading it. I have only a few minor comments and typos to point out.

**Specific comments**

L. 153: It is unusual to introduce a figure by referring only to subfigure (b). Consider swapping (a) and (b). Also, it would be helpful to include the all-time average of monthly precipitation frequencies for daily >0 mm and daily >1 mm, in addition to the reported range.

L. 274: The word "stick" is a bit informal in scientific writing. Suggested revision: "we rely on the corrected Pluvio data."

Figure 6: The legend does not fully represent the figure. The black and dark gray bars appear with white fill inside. Please update the legend accordingly.

Table 3 caption: "the TS method" should likely be "the TMS method."

---

## Author Response (AR2)

We thank the two anonymous reviewers for their time in reviewing the revised version of the manuscript. In the following, the answers to the reviewers' comments and questions are given in red. Line numbers, if not stated otherwise, refer to the original version of the manuscript and may have changed in the revised version.

**Reviewer #1:**

General comments

The authors have substantially improved the manuscript by removing the section on the MRR, reorganizing the text, clarifying the focus, and improving the figures. The responses to the reviewer comments are also excellent. I really appreciate the careful work by the authors, and I am pleased with the changes made to the manuscript. The paper now reads very well, and I truly enjoyed reading it. I have only a few minor comments and typos to point out.

Specific comments

L. 153: It is unusual to introduce a figure by referring only to subfigure (b). Consider swapping (a) and (b). Also, it would be helpful to include the all-time average of monthly precipitation frequencies for daily >0 mm and daily >1 mm, in addition to the reported range.

We swapped the sub-figures a) and b) and also added the all-time average values for the monthly precipitation frequencies for daily >0 mm and daily >1 mm in the text.

L. 274: The word "stick" is a bit informal in scientific writing. Suggested revision: "we rely on thecorrected Pluvio data."

changed

Figure 6: The legend does not fully represent the figure. The black and dark gray bars appear with white fill inside. Please update the legend accordingly.

The legend has been updated.

Table 3 caption: "the TS method" should likely be "the TMS method."

changed

**Reviewer #2:**

The author addressed most of the comments and I really thank them for the effort they put in this work. I think the remaining issue is the lack for a real discussion on your results. For example the separation between solid and liquid precipitation is close to 2°c, which is very high compared to other studies taking frequently 1°c. You need to explain why it is that high and to show the implications for other studies. Is the snow falling between 0 and 2°c really able to stick the ground? (which is the main interest for most studies).

Please see the answer to this comment with respect to the comment on l.314

You also do not really discuss previous studies on linking precipitation and atmospheric circulation. What other studies have shown? How your study improve what was previously shown?

We actually present different studies regarding the impact of atmospheric circulation on precipitation at Svalbard (lines 71-88 in the new version). To better highlight this topic, we separated the paragraph in the introduction from the discussion on extreme precipitation events. We also shifted the findings from Serreze et al. (2015) from the results part to the introduction. Until now, no study has specifically associated atmospheric rivers, cyclones, and fronts with Ny-Ålesund precipitation, to which we could compare our results. A direct comparison with the study by Lauer et al. (2023) is difficult since it focuses on only two shorter time periods and larger Arctic domains. We comment on this in lines (new version ll. 408-411).

However, concerning extreme precipitation in Svalbard, our results are in line with the findings of Serreze et al. (2015): we see enhanced water vapor transport with pressure patterns favoring water vapor transport from the lower latitudes. We comment on this in l. 465-467 (new version). Serreze et al. (2015) did not specifically look into specific weather systems and also did not quantify the relative contribution of these systems to precipitation.

For the method I also have one concern: Why you don't compare your results with the Wolff method used in Champagne et al.? Using the ensemble mean correction introduced a bias between methods, so it is hard to separate the origins of the error here.

Please see the answer to the comment with respect to the comment on l.262

Here are some specific comments:
l.12 : you mean days with highest daily rate ?

Yes. We changed the sentence to:

"Extreme events, defined as days with daily precipitation sums above the 98$^{th}$ percentile, contribute 18% to the total precipitation amount."

l.13 : with fronts and high liquid mass fraction

changed

l.16 : comma between variable and crucial ?

We rephrased the sentence to:
"Precipitation is a key climate variable that is critical to the Arctic climate system."

l.30 : lapse rate of what ?

It is the temperature lapse rate. With regard to Arctic amplification, "lapse rate feedback" is a common name. We thus stick to this wording.

See, for example
Linke, O., Quaas, J., Baumer, F., Becker, S., Chylik, J., Dahlke, S., Ehrlich, A., Handorf, D., Jacobi, C., Kalesse-Los, H., Lelli, L., Mehrdad, S., Neggers, R. A. J., Riebold, J., Saavedra Garfias, P., Schnierstein, N., Shupe, M. D., Smith, C., Spreen, G., Verneuil, B., Vinjamuri, K. S., Vountas, M., and Wendisch, M.: Constraints on simulated past Arctic amplification and lapse rate feedback from observations, Atmos. Chem. Phys., 23, 9963–9992, https://doi.org/10.5194/acp-23-9963-2023, 2023.

l.39 : trend of precipitation

changed

l.59.60 : I don't understand how increase in inter-annual variability suggest that extreme precipitation is becoming more likely. Explain.

Since both, the mean of precipitation and its variability increase, extreme precipitation events are more likely. Bintanja et al. (2020) explicitly state in their abstract: "Because both the means and variability of Arctic precipitation will increase, years/seasons with excessive precipitation will occur more often, as will the associated impacts." Please see also the discussion section in Bintanja et al. (2020): "Increased precipitation variability on top of rising mean precipitation rates can potentially exert severe consequences (10), since both increase the likelihood of wet extremes (21) with large and possibly irreversible hydrological/ecological (e.g., water availability, marine productivity, and permafrost thaw), societal (e.g., local communities), and economic (e.g., infrastructural damage) impacts (10, 22–26). Extremely wet episodes are thus likely to become far more common in the Arctic's (near) future; the unusually wet autumn/winter of 2015/2016 and 2016/2017 in Svalbard (causing a number of climate refugees to abandon their homes) may already have signaled the emergence of extreme Arctic precipitation events along with their long-lasting impacts."

l.58-82 : I think this paragraph would fit better before the previous one to increase the flow of idea.

We thought of about changing the order of the paragraphs. However, since the importance of precipitation phase is highlighted in the lines 33-47, the paragraph on the discrimination phase comes naturally afterwards. We can see that different ways of the logical flow are possible but we would like to stick to the current one.

l.89:94 : maybe it is a bit too detailed here. Some details can be in method section.

We mention these details here because this paragraph also highlights what is new about this data set compared to other studies using classical manual precipitation gauges.

l.145 :10 or 20 %?

The catch efficiency and, thus also the improvement of using a windshield depends on the type of the shield, type of precipitation, and wind speed. This is why the range of 0.1 and 0.2 is reported here.

l.151 : you can remove « made a choice here but want to »

removed "made a choice here"

l.151-153 not sure this is needed. Also in fig.2 why showing minutes and hourly resolution in the same graph ? It is a bit confusing. If parsivel is 1min resolution you can maybe put the pluvio minute resolution in the upper panel.

We assume that you are referring to lines 153-155 (old manucript). We swapped panels a) and b) as suggested by reviewer 1.  Panel a) shows Pluvio data and Panel b) shows Parsivel data. For the Pluvio, we also report results in 1 min and daily resolution because often, precipitation data are only daily resolved. We think that this is useful to better understand the data set and the impact of data sampling.

l.183 : you mean from June to September ?

Yes, changed.

l.260-261 : where is the wind measured ? This can greatly impact the correction. It can be also the pluvio that gives better results. If the wind is measured in a open area, it would be stronger and could lead to an overcorrection of MET Norway data.

Champagne et al. (2024) used MET Norway wind speed measurements. The location of these measurements changed over time (see Fig. S1 in the Supplementary Material of Champagne et al (2024); https://doi.org/10.1175/JHM-D-23-0182.s1). For the overlapping time period with the Pluvio data, Champagne et al. (2024) took 10 m wind speed measurements from a Vaisala WAA 151 in the open measurement field (see Fig. S1 in the Supplementary Material by Champagne et al.). This location is about 160 m away from the BSRN station wind sensor that we used to correct the Pluvio data (see Fig. R1).

[Figure]

*Figure R1: Location of Pluvio, MET Norway precipitation gauge and wind sensors. Google Maps.*

Champagne et al. (2024) calculated the wind speed at gauge height (2 m) from the 10 m height wind speed measurements (Eq. 1 in Champagne et al. (2024)), assuming a roughness length of 0.02 and an average vertical angle of obstacles around the gauge of 12. This wind

speed estimate is very uncertain; on the one hand, due to the logarithmic extrapolation to 2 m, on the other hand, due to the distance to the MET Norway precipitation gauge. Even though they tried to include effects of the surrounding buildings on the wind speed, the wind speed estimate is still quite uncertain. For the Pluvio measurements, we use wind sensor measurements at the same height, which are only about 40 m away from the Pluvio. So these wind speed measurements should be a very good estimate for the actual wind speed at the Pluvio.

Furthermore, Champagne et al. (2024) used wind speed measurements at 0600, 1200, and 1800 UTC to correct the precipitation measurements while we used minute resolved wind speed and precipitation data, resulting in a better temporal matching, i.e. using the actual wind speed at the time when precipitation was observed. All these aspects might result in the observed differences between the corrected precipitation data from Pluvio and the MET Norway gauge.

We expanded the discussion about the differences between der Pluvio and MET Norway data sets in the manuscript (new version ll. 259-303) and also adapted the outlook section (new version ll. 538 ff).

l.262 : I don't know if it's because of the Wolff method. It can be as you said that the wind is larger at the pluvio site than at the MET Norway site.

If we just compare our results to the corrected precipitation data of Champagne et al. (2024) using the Wolff et al. (2015) only method with 2m wind speed, the differences are even slightly larger (see Fig. R2).

[Figure]

*Figure R2: Scatter plots of monthly (left) and daily (right) precipitation sums at Ny-Ålesund for 1 August 2017 – 31 December 2021. Corrected monthly precipitation amount of MET Norway precipitation gauge (Wolff et al. (2015) correction from Champagne et al., 2024) vs. Pluvio (with Wolff et al. (2015) correction).*

So the differences in the Pluvio and MET Norway precipitation data sets are not only due to the usage of different corrections functions but likely also related to the different temporal resolutions/data sampling (1 min vs. 12 h resolved data, sampling of T and wind speed data), different T and wind speed data sets, and differences in how the wind affects the

measurements due to the different locations. On the one hand, it might be that the wind effect is overestimated in Champagne et al. (2024), on the other hand, the undercatch correction of Pluvio might be too small. Different data sampling and different temporal resolutions of the data sets also have quite some impact on the corrected precipitation amount. As mentioned in the manuscript, Jacobi et al. (2019) resampled the high-resolution precipitation data by Pluvio and Geonor to 1 h and 24 h intervals, respectively. For the temporally coarser resolved data, the correction was much larger, no matter which correction function was applied (see Table R1).

*Table R1: Accumulated precipitation at Ny-Ålesund covering a full hydrological year from Sep 2017 to Sep 2018 based on three different precipitation sensors and different corrections functions( Førland and Hanssen-Bauer (2000); Wolff et al. (2015)) applied to different temporally resolutions (from Jacobi et al, 2019).*
*Førland, E.J., Hanssen-Bauer, I. Increased Precipitation in the Norwegian Arctic: True or False?. Climatic Change 46, 485–509 (2000). https://doi.org/10.1023/A:1005613304674*

**Annual accumulation 2017-2018**

| | | Correction according to | | | |
|---|---|---|---|---|---|
| | | F & H-B (2000) | | Wolff (2015) | |
| Accumulation | Observed | 24-hr | 1-hr | 24-hr | 1-hr |
| Manual | 657 | 817 | ·/· | 790 | ·/· |
| Pluvio | 589 | 770 | 705 | 731 | 673 |
| Geonor | 588 | 781 | 709 | 748 | 670 |

Yearly precipitation sums increased by about 70–80 mm when the 24 h resolved data were used.

In the present study, we cannot fully explain the differences between the data sets. A more detailed comparison, also between Geonor and Pluvio, which are installed in the same field, is needed to gain a better insight into the uncertaintes.

We expanded the discussion about the differences between der Pluvio and MET Norway data sets in the manuscript (new version ll. 259-303) and also adapted the outlook section (new version ll. 538 ff).

l.264-l265 : the absolute difference is simply because corrected precipitation are higher than uncorrected ?

We removed this sentence from the manuscript and expanded the discussion on the differences (new version ll. 259-303).

l.314 : 1.8°c ? It seems high compared to other studies ! You need to discuss on that.

In Champagne et al. (2024), a 1°C threshold is used to separate snow and rainfall. They do not further comment on why they chose that threshold but in the conference contribution by Jacobi et al. (2019), a similar threshold has been presented (see Fig. R3)

[Figure]

*Figure R3: Solid and liquid fraction for the daily precipitation as a function of daily mean temperature. The data was derived using visual observations and recorded temperatures from the period 1975-2007 (from Jacobi et al., 2019).*

They show that solid and liquid precipitation equally occur at around 1°C±0.5°C. Their analysis is based on 24h accumulated precipitation measurements, 24h averaged temperatures and weather observers' reports of precipitation type. The latter is of course subjective. Furthermore, the reported 24h mean temperature is not the actual temperature at which precipitation occurred.  They also considered a different time period (1975-2007). This is why the solid/liquid fraction as a function of temperature likely differs.

We added a discussion on this in ll. 342-349.

l.316-325 : It is not clear to me what is the temperature based mass separation ? You simply use the % mass fraction per temperature that was derived from parsivel?

Yes. This is exactly how it is written in the manuscript. We explicitly refer also to Fig. 5 b.

Also you used the corrected precipitation data in the calculation of liquid precipitation ? You need to clarify here.

Yes, we use the corrected precipitation data for all analyses in section 3.2 and the following. We mention this at the end of section 3.1: "For the following analyses, we rely on the corrected Pluvio data using the Wolff et al. (2015) method."

We changed the sentence to:
"To split the corrected Pluvio precipitation amount into solid and liquid for the whole period August 2017 to December 2021, …"

We also changed the captions of Tables 3 and 4 and Figs. 8 and 11 to make clear that the results are based on corrected Pluvio data.

Figure 6 : I don't see dark grey bars, I see white bars. What are these white bars representing? Table 4 : wouldn't be better to write 'no system' instead of residual?

We changed the caption to:
"Figure 6. a) Total monthly precipitation (in mm) from corrected Pluvio data (black contour bars). The corresponding liquid precipitation amount (in mm) from the combined Parsivel/temperature-based mass separation (Parsivel/TS; filled red bars) and the monthly liquid fraction (in %, dotted line) are shown as well. b) Differences in monthly liquid

precipitation amount (in mm) if the temperature-based mass separation (TMS; dark gray contour bars) or a simple temperature threshold of $1\circ$C (T1$\circ$C; filled light gray bars) is used."

In Lauer et al. (2023), precipitation amount that could not be attributed to a weather system was called "residual". We thus use the same term.

337-341 : I don't understand what you did here

The question is if differences in the used temperature thresholds in different studies are due to different data sampling, i.e. 1 min in our case and 1 h (as for example, used in Champagne et al. (2024)). Thus, we wondered if resampling our highly resolved data to hourly data would result in a different liquid/solid fraction-temperature relationship. However, this is not the case.

l.355 : you should talk about the average before talking about the monthly average.

We swapped the order of the sentences:

"On average, fronts occur 14% of the time at Ny-Ålesund. Monthly front occurrence (separated and co-located) shows maxima of more than 20% in summer or late summer."

l.363 : precipitation amount ?

We changed this sentence to:

"….to the total precipitation amount from Aug 2017 to Dec 2021."

l.372 : here you should talk about the long term average before the specific months.

This is unclear to us. In l. 372, we talk about the long-term average:
"Regarding the whole time period, separated fronts contribute only about 4% to the total precipitation. "

We prefer to keep the paragraph as it is.

l.384-385 : AR has a higher occurrence in summer as well

We had a closer look into the data again and a substantial amount of precipitation for both ARs and fronts falls in the warmer months May to September. Specifically, this is 49% of the precipitation associated to ARs and 56% of the precipitation related to fronts.

We thus rephrased the sentence to:
"The high liquid fraction of precipitation related to ARs and fronts is also due to the fact that a substantial amount of precipitation associated with these weather systems, i.e. 49% for ARs and 56% for fronts, falls in the warmer months May to September."

We also added numbers on the frequency of occurrence of fronts and ARs in summer vs. other months when presenting Fig. 7.

"A seasonal dependency is not clearly evident from this short period, although the occurrence of ARs is slightly higher on average in summer (12%) than in the other months of the year (7%)."

and

"On average, front occurrence in June, July and August is 24% compared to 10% during the other months of the year."

l.389-392 : what is the value of these graphs for your analysis? You don't compare hourly with daily here.

To our knowledge, hourly precipitation rates have not been characterized yet for Ny-Ålesund. This is of high interest not only for process studies but also for model comparisons in the future. In addition to total (monthly, yearly) precipitation, hourly (and daily) precipitation rates are relevant for the hydrological cycle. Before discussing the impact of the weather systems on precipitation on hourly and daily scales, we would like to present the general characteristics of precipitation at Ny-Ålesund first. What are typical hourly/daily precipitation sums? Which events would be regarded as extreme events? We do not intend to compare hourly with daily data.

l.395 : « Hourly liquid precipitation amounts are typically between 0.1–1.0 mm (25th and 75th percentiles) ». I am not sure this is needed.

We prefer to keep this sentence.

l.389-402 : I am wondering how much of the higher liquid precipitation with AR and FR is due to higher occurrence of AR and FR in summer.

See answer to previous comment.

l.402 : maybe it needs two digits here. It may be 0.04 and not 0?

It is 0~mm. Since there are many no-liquid precipitation cases in the residual class, the median value is indeed 0. This is why we didn't include a further digit.

l.403-419 : I don't really understand how this part is related to what you talked about right before. I think it needs to be later in the manuscript (in a discussion part?) and needs clarity on what is the purpose of this discussion part.

The high-resolution Pluvio measurements make it possible to assess the amount of precipitation on different temporal scales (instead of just characterizing daily or monthly precipitation data).  Section 4 presents the impact of different weather systems on precipitation amount and type at Ny-Ålesund. To get an overall picture, we focus first on the monthly and whole-time characteristics before zooming into smaller time intervals, i.e. hourly and daily.

Looking just at the monthly data does not give us information about the timing of the precipitation. In addition to the monthly precipitation amount, the precipitation rates on shorter time scales are important.
We added a sentence at the beginning of section 4 to clarify this flow of thoughts.

"We first have a look at the monthly and whole-time statistics before zooming into hourly and daily precipitation data. As outlined in section 1, not only the total precipitation amount but also the precipitation intensity is a decisive variable for the Arctic climate system"

l. 421-422 : where do you find that ?

We changed the sentence to:

"When focusing on the right tail of the distribution of the daily precipitation amounts, in particular on the 2% of the days with the highest precipitation amounts (Table 6), we find from inspection of ERA5 reanalysis data (not shown) that all of these events are related to enhanced water vapor transport from the North Atlantic or Eurasia, often in the form of ARs and in combination with fronts."

l.443 I wouldn't say they are common. I would say they could occur.

changed

l.459 : « The temperature dependency of the mass separation follows the temperature relation of the phase occurrence ». not very clear, reformulate.

We rewrote the sentence:

"The temperature dependence of liquid/solid mass separation is similar to the temperature dependence of liquid/solid precipitation occurrence."

l.460 : you could say this suggest that hourly resolution is enough for phase separation.

Applying a temperature-based mass separation method will still result in uncertainties for both minute-resolved and hourly-resolved precipitation data. So when there is Parsivel information in high-temporal resolution, also precipitation data in high temporal resolution is preferred. So, we would not generally say that hourly resolution is enough for phase separation.

l.464 : 6 to 15 % depending on the years ?

Yes, we mentioned "annual liquid precipitation sums".

Your conclusion is maybe a bit too much detailed. It needs to point out your significant results and the novelty of your work compared to previous studies.

We believe that the main points are presented in a concise way. The summary and conclusions section is divided into 7 paragraphs following the logical order of the manuscript:

1) Outline of the study with instrument setup highlighting the advantage of the new measurements as well as weather system analysis which has not yet been applied to Ny-Ålesund data
2) Main outcomes of phase discrimination analysis using Parsivel and temperature data
3) Discussion on the limitation of Parsivel measurements and outlook for future validation of Parsivel measurements
4) Main outcomes of impact analysis of weather systems on precipitation at Ny-Ålesund (total amount, monthly amount, and hourly intensities)
5) Main outcomes of daily precipitation amount analysis incl. trace precipitation and extreme events
6) Discussion on uncertainties of precipitation estimates and outlook on future comparison studies
7) Discussion on weather system definition and outlook in future usage of auxiliary data sets

Paragraph 3) could have been shifted also to the end, but we found it easier to follow the logical flow if this is discussed directly after the presentation of the corresponding main results.

 Overall the manuscript has been greatly improved.

Thank you very much. We really appreciate your detailed comments.